# Utilizing probability estimates from machine learning and pollen to understand the depositional influences on branched GDGT in wetlands, peatlands, and lakes

Amy Cromartie[1], Cindy De Jonge[2], Guillemette Ménot[3], Mary Robles[4,5], Lucas Dugerdil[3,5], Odile Peyron[5], Marta Rodrigo-Gámiz[6], Jon Camuera[7], Maria Jose Ramos-Roman[8], Gonzalo Jiménez-Moreno[6], Claude Colombié[9], Lilit Sahakyan[10], and Sébastien Joannin[5]

[1]Université Côte d'Azur, CNRS, CEPAM, UMR 7264, 06300 Nice, France

[2]Geological Institute, ETH Zürich, 8092 Zurich, Switzerland

[3]ENS de Lyon, Université Lyon 1, CNRS, UMR 5276 LGL-TPE, 69364 Lyon, France

[4]Aix-Marseille Univ., CNRS, IRD, INRAE, Coll France, UMR 34 CEREGE, 13545 Aix-en-Provence, France

[5]ISEM, Univ. Montpellier, CNRS, IRD, 34090 Montpellier, France

[6]Department of Stratigraphy and Paleontology, University of Granada, 18071 Granada, Spain

[7]Unit of Botany, Faculty of Pharmacy, Complutense University of Madrid, 28040 Madrid, Spain

[8]Instituto de Investigación en Cambio Global Universidad Rey Juan Carlos 28933, Madrid, Spain

[9]Univ Lyon, UCBL, ENSL, UJM, CNRS, LGL-TPE, Villeurbanne, F-69622 France

[10]Institute of Geological Sciences, National Academy of Sciences of Republic of Armenia, Yerevan 0019, Armenia

**Correspondence:** Amy Cromartie (aec277@cornell.edu)

**Abstract.** Branched glycerol dialkyl glycerol tetraethers (brGDGTs) are critical molecular biomarkers for the quantitative reconstruction of past environments, ambient temperature, and pH across various archives. However, numerous issues persist that limit their application. The distribution of brGDGTs varies significantly based on provenance, resulting in biases in environmental reconstructions that rely on fractional abundances and derived indices, such as $MBT'_{5ME}$. This issue is especially significant in shallow lakes, wetlands, and peatlands, where ecosystems are sensitive to diverse environmental and climatic factors. Recent advancements, such as machine learning techniques, have been developed to identify changes in provenance; however, these techniques are insufficient for detecting mixed environments. The probability estimates derived from five machine learning algorithms are employed here to detect provenance changes in brGDGT downcore records and to identify periods of mixed provenance. A new global modern database ($n = 2031$ TSI) was compiled to train, validate, test, and apply these algorithms to two sedimentary records. Our findings are corroborated by pollen, non-pollen palynomorphs, and X-ray fluorescence (XRF) obtained from the same sedimentary core sequence. These microfossil and geochemical proxies are utilized to discuss changes in provenance, hydrology, and ecology that influence brGDGT provenance. Probability estimates derived from random forest with a sigmoid calibration are most effective in detecting changes in brGDGT provenance. Minor changes in the relative contributions of brGDGT provenance can significantly influence the distribution of brGDGT, especially regarding the $MBT'_{5ME}$ index.

## 1 Introduction

Branched glycerol dialkyl glycerol tetraethers (brGDGTs), first identified in peat sequences (Weijers et al., 2006), have demonstrated significant potential as a quantitative proxy for paleoenvironmental reconstructions. The ubiquity of brGDGTs and their global correlations with tempera-

ture and pH, notably across different archive types, positions them as a valuable tool for paleoclimate reconstructions (among others, Weijers et al., 2007; Peterse et al., 2012; Loomis et al., 2011; Raberg et al., 2022a, b). Researchers have identified brGDGTs across various depositional environments, such as peat, soils, loess, and fossilized bones and lacustrine, marine, and river sediments (e.g., Weijers et al., 2006, 2007; De Jonge et al., 2014a; Warden et al., 2016; Naafs et al., 2017a, b; Dillon et al., 2018; Baker et al., 2019) at differing geological timescales, indicating their widespread potential as a proxy for reconstructing the continental paleoclimate.

A key challenge in utilizing brGDGT-based reconstructions in continental settings is the temperature-independent variability in fractional abundance (FA) distribution across these environments (De Jonge et al., 2014b; Naafs et al., 2017b; Dearing Crampton-Flood et al., 2020; Martínez-Sosa et al., 2021; Raberg et al., 2022b). In the context of lacustrine, wetland, and peat archives, the fractional abundance of brGDGTs produced in aquatic environments and surrounding soils varies (Tierney and Russell, 2009; Tierney et al., 2010; Zink et al., 2010; Buckles et al., 2014; Loomis et al., 2011, 2012, 2014a, b; Li et al., 2016; Russell et al., 2018; Dang et al., 2018). Potential changes in provenance thus result in distribution differences that may lead to inaccuracies in paleoenvironmental reconstructions. This includes paleotemperature reconstructions derived from the widely recognized index based on the methylation of branched tetraether (MBT) of 5-methyl ($MBT'_{5ME}$). This index measures the degree of methylation of the 5-methyl brGDGTs, distinguishing it from the 6-methyl brGDGTs to establish calibrations that exhibit a stronger correlation with mean annual air temperature (MAAT) (De Jonge et al., 2014a). The $MBT'_{5ME}$ index has been successfully utilized as grounds for various global temperature calibrations because of its strong correlation to temperature in modern samples that include lakes, peats, and soils (e.g., De Jonge et al., 2014a; Hopmans et al., 2016; Naafs et al., 2017a; Dearing Crampton-Flood et al., 2020; Martínez-Sosa et al., 2021; Véquaud et al., 2022). Provenance changes may introduce bias to temperature reconstructions based on the $MBT'_{5ME}$ index due to its value generally being higher in soils than in lakes (Martínez-Sosa et al., 2021).

Furthermore, brGDGT distributions in these depositional environments may be influenced by distinct environmental characteristics. Soil chemistry, particularly pH, can influence the 5-methyl brGDGTs (De Jonge et al., 2021, 2024). In certain lakes, the 6-methyl brGDGTs exhibit a stronger correlation with mean annual air temperature compared to the 5-methyl brGDGTs, which contrasts with the catchment soils (Dang et al., 2018). In peatlands, the $MBT'$ and $MBT'_{5ME}$ values are higher in dry sites compared to those that are waterlogged (Rao et al., 2022). Factors influencing the distribution of brGDGTs in lakes include lake stratification and redox conditions (Weber et al., 2018), salinity (Wang et

al., 2021), conductivity (Tierney et al., 2010; Raberg et al., 2022b), dissolved oxygen (Wu et al., 2021), and water depth (Stefanescu et al., 2021), amongst others. In soils, vegetation and vegetation-mediated factors such as soil temperature (Liang et al., 2019, 2023), soil moisture (Menges et al., 2014; Dang et al., 2016), precipitation (Dugerdil et al., 2021a), and soil chemistry (Dang et al., 2016; De Jonge et al., 2021) all influence distributional changes. BrGDGT distributions in peat may vary in response to flooding, the drying of peatlands, and alterations in the water table (Rao et al., 2022; Ofiti et al., 2024). The potential differential distributions resulting from depositional environments underscore the influence of changes in provenance or hydrological conditions on brGDGT-based environmental reconstructions.

BrGDGT reconstruction in Quaternary downcore lacustrine records indicates that changes in depositional and mixed provenance significantly affect environmental reconstructions (i.e., Martin et al., 2019; Robles et al., 2022; Ramos-Román et al., 2022; d'Oliveira et al., 2023; Acharya et al., 2023). As climatic or successional changes occur concurrently with temperature variations, isolating the effects of provenance changes on $MBT'_{5ME}$ is challenging. Several indexes and ratios have been developed to detect brGDGT provenance change. The BIT index (Hopmans et al., 2004), and later the IIIa / IIa ratio (Xiao et al., 2016), for example, were designed to identify terrestrial organic input in marine sediments. Although useful in marine contexts, these indexes have had limited success in lacustrine terrestrial environments (e.g., Martin et al., 2020). Ternary diagrams are commonly used to visualize brGDGT (e.g., Russell et al., 2018), enabling a comparison between fossil and modern datasets. These diagrams, however, reduce the data size to three variables, limiting their usefulness in isolating the influence of provenance change on the individual brGDGT isomers. Recently, Martínez-Sosa et al. (2023) employed supervised machine learning (ML) to identify changes in provenance using classification models based on modern samples. Their success highlights the power that ML applications can have in solving difficult issues. ML applications differ from traditional statistics applications by focusing on prediction rather than inference (Bzdok et al., 2018). ML's power over these conventional methods lies in its ability to handle data with multiple variables for a few subjects while examining nonlinear relationships within the datasets (Bzdok et al., 2018). The models of Martínez-Sosa et al. (2023) proved effective at identifying shifts in provenance, but a limitation of their study, however, is the inability to detect periods of mixed provenance.

This paper aims to correct that by introducing a strategy for identifying provenance changes across depositional lacustrine, peat, and soil environments, including mixed contexts, utilizing a new global brGDGT database and ML techniques, as well as environmental reconstructions based on pollen, non-pollen palynomorphs (NPPs), and X-ray fluo-

rescence (XRF) datasets. Two approaches are employed to achieve this objective.

First, we use probability estimates derived from ML to identify changes in provenance over time, extending the work of Martínez-Sosa et al. (2023). Rather than employing discrete classification, as they did, we utilize the probability estimates from these classification algorithms to analyze the contributions from differential provenance at any specific time. This method enhances prior approaches by recognizing environments that integrate brGDGTs from multiple inputs and depositional settings – and thus multiple provenances – that have not fully transitioned to a new depositional state. The probability estimates are derived from the classification of modern samples ($n = 2301$), categorized into three groups: soil, peat, and lake, utilizing both previously published and new datasets. We test five popular parametric and non-parametric ML models based on their ability to handle small tabular datasets and produce reliable probability estimates when calibrated (Malley et al., 2012; Wang et al., 2019). Models utilizing different structures were chosen, including simple tree-based algorithms (classification and regression trees), ensemble trees (random forests), linear models (logistic regression), margin-based classifiers (support vector machines), and instance-based lazy learners ($K$-nearest neighbors) to evaluate performance. The best-performing model was then chosen to apply to two downcore sedimentary sequences. These are employed using Python and scikit-learn to identify intervals where downcore records are predominantly influenced by in situ lake brGDGTs, mineral soils, and peatlands, as well as combinations of these elements.

Second, to ensure the accurate identification of provenance changes, comparisons are conducted with published pollen, NPPs, and XRF from extensive sediment records and variations in brGDGT distribution (i.e., Robles et al., 2022; Camuera et al., 2018, 2019; Ramos-Román et al., 2018; Rodrigo-Gámiz et al., 2022). The records are situated in the semi-arid mid-latitude zones, where water bodies are subject to temporal variations. Aquatic pollen and NPPs have previously been used to verify changes in provenance in brGDGT communities from fossil records (i.e., Robles et al., 2022; d'Oliveira et al., 2023; Ramos-Román et al., 2022; Barhoumi et al., 2023). In addition, we also compare our results with XRF core scanning data from the same sedimentary sequence. Utilizing these proxies allows for an independent comparison of outputs to (i) confirm ML results through the integration of brGDGT-based reconstructions with pollen, NPPs, and XRF and (ii) demonstrate how these complementary proxies can aid in identifying potential hydrological, ecological, and depositional changes that may cause provenance shifts, thus introducing bias in brGDGT reconstructions. This study demonstrates that alterations in provenance and hydrology can significantly influence the distribution of brGDGTs and, consequently, establish indices

like $MBT'_{5ME}$, while also offering novel methodologies for identifying changes in global paleorecords.

## 2 Materials and methods

### 2.1 GDGT databases

#### 2.1.1 Building a new modern sample database

This study compiles published brGDGT databases for depositional lake ($n = 591$), soil ($n = 1197$), and peat ($n = 532$) categories (Baxter et al., 2019; Cao et al., 2020; Chen et al., 2021; Dang et al., 2018; De Jonge et al., 2014b; Dearing Crampton-Flood, 2020; Dearing Crampton-Flood et al., 2020; Ding et al., 2015; Dugerdil et al., 2021a, b; Guo et al., 2020a, b; Halffman et al., 2022; Jaeschke, 2018; Jaeschke et al., 2018; Kirkels et al., 2020; Kou et al., 2022; Li et al., 2017; Liu et al., 2020; Martin et al., 2019, 2020; Martínez-Sosa et al., 2021; Naafs, 2017; Naafs et al., 2017a, b; Ning et al., 2019; Pérez-Angel et al., 2020; Qian et al., 2019; Raberg et al., 2021a, b; Rao et al., 2020, 2022; Robles et al., 2022; Russell et al., 2018; Sinninghe Damste et al., 2020; Stefanescu et al., 2020, 2021; Véquaud et al., 2021b, a; Wang et al., 2016, 2020a, 2018, 2020b; Wang and Liu, 2021; Weber et al., 2018; Wu et al., 2021; Xiao et al., 2015; Yang, 2020; Yang et al., 2015; Yao et al., 2020; Fig. 1, full data at https://doi.org/10.17632/tr8tppy9fz.1, Cromartie, 2025a). Round robin test results show that results from multiple laboratories can be integrated into a single database (De Jonge et al., 2024). Results were included only when chromatography enabled the separate quantification of 5- and 6-methyl brGDGTs (i.e., De Jonge et al., 2014b). The fractional abundances of 15 distinct brGDGT structural isomers were sourced from the original authors or recalculated from the initial datasets. We enhanced the training dataset for certain published datasets by obtaining data with greater precision from the original authors, where fractional abundances had been rounded to two decimal places. We incorporated the fractional abundances of individual downcore samples from Naafs et al. (2017a) to enhance the sample size of our peat analysis. This facilitated the development of a more robust model for assessing brGDGT distribution across various types. All samples originate from terrestrial environments (Fig. 1). The 7-methyl (Ding et al., 2016) or the 5/6 isomer, also referred to as IIIa″ (Weber et al., 2015), were excluded due to limited data. The original authors' description was utilized to categorize the samples into a classification index (i.e., soil, lake, peat). Suspended particulate matter (SPM), moss pollsters, marine, and river samples were excluded. Latitude and longitude data were converted to decimal degrees as required. The Köppen–Geiger classification of each modern sample was done with the kgcpy library (Yu et al., 2024) in Python to assess the climate distribution.

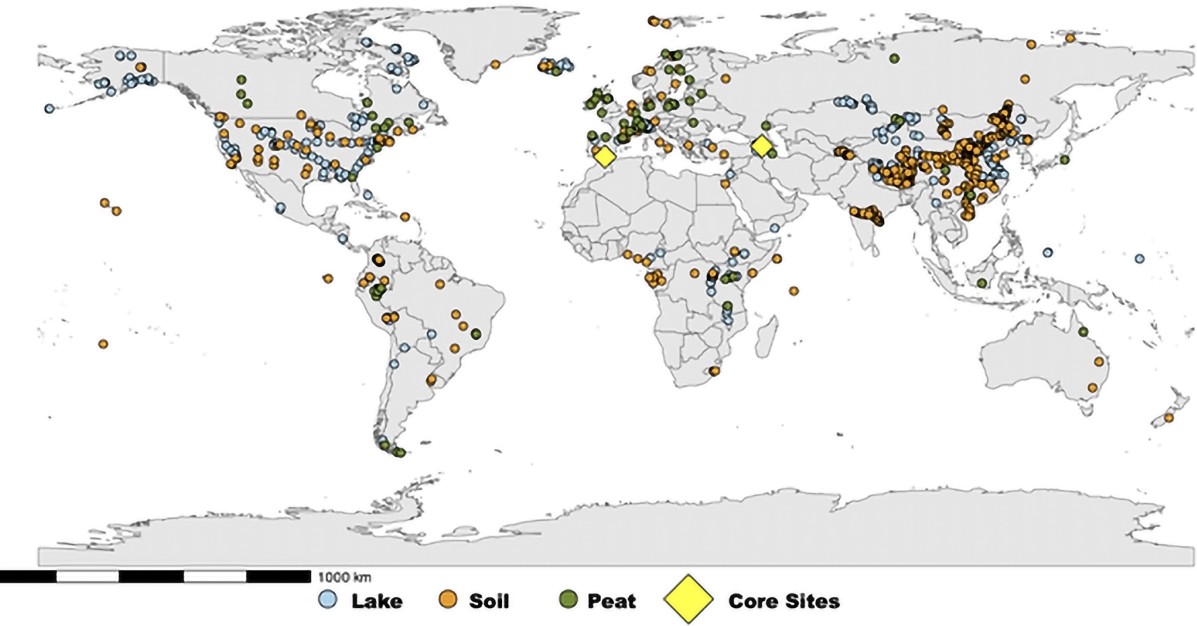

**Figure 1.** Map of modern sample locations used in the compiled database alongside the two sites designated for paleo-reconstructions. Map created with R package ggplot2 (Wickham, 2016).

### 2.1.2 Addition of new samples from Armenia

A total of 30 new surface samples from the country of Armenia were added to the global dataset to expand the database for semi-arid environments. Nine samples were collected from wetlands at a depth of 0–2 cm; one sample was collected from Lake Sevan at a depth of 2–3 cm, as previously discussed in Robles et al. (2022); and 20 surface soil samples were collected. For brGDGT extraction, each sample was first lyophilized, freeze-dried, and ground. Lipids were extracted from 0.5 to 1 g of sample in two rounds with a MARS 6 CEM microwave, using a 3 : 1 mixture of dichloromethane (DCM) and methanol (MeOH) (3 : 1). These samples were filtered with a silicon SPE cartridge with a mixture of hexane : DCM (1 : 1) and then DCM : MeOH (1 : 1) to separate the apolar and polar fraction, respectively. An internal standard of $C_{46}$, following Huguet et al. (2006), was added to the total lipid extract (TLE) prior to separating the fraction. The polar fraction was then analyzed on high-performance liquid chromatography with atmospheric pressure chemical ionization mass spectrometry (HPLC-APCI-MS, Agilent 1200) at LGL-TPE ENS, which allows separation of the 5- and 6-methyl GDGTs, following Hopmans et al. (2016). Selective ion monitoring (SIM) of $m/z$ of 1050, 1048, 1046, 1036, 1032, 1022, 1020, 1018, and 744 was used for the brGDGT isomers and the internal $C_{46}$ standard (Hopmans et al., 2016; Davtian et al., 2018; Huguet et al., 2006).

### 2.1.3 Resampling and balancing modern dataset

The distribution of modern brGDGT samples across classification datasets (i.e., soils, lakes, peats) was not uniform, with a predominance of soil samples and an underrepresentation of lake and peat samples (Fig. 1). Unbalanced datasets can result in considerable performance issues, such as the misclassification of data with limited sample sizes, which may prevent the learning algorithm from identifying general patterns within the datasets (He and Garcia, 2009). Consequently, a combination of downsampling and upsampling techniques was utilized for model comparisons (Fig. 2). This involved evaluating each ML model using both the raw and resampled datasets, which incorporated upsampled synthetic samples. In the resampled dataset, we initially preformed random downsampling of the soil samples in R to achieve a sample size of 750 from the original dataset. The synthetic minority oversampling technique (SMOTE) function from the R library smotefamily (Siriseriwan, 2019) was employed to upsample the peat and lake datasets. The SMOTE function is an oversampling technique that selects a sample from the minority dataset, identifies its nearest neighbor(s), and generates a new data point between the original pair (Siriseriwan, 2019). SMOTE was utilized to generate 1000 synthetic samples for the lake and peat datasets derived from the original datasets. The distribution of the SMOTE samples and original database were plotted, and a principal component analysis and Kolmogorov–Smirnov test were run to verify that no bias was introduced (results in Figs. S3 and S4 and Table S1 in the Supplement). Samples were randomly selected from

the synthetic dataset to adjust the raw datasets for lake and peat to a total of 750. In the case of the peat and lake samples, 219 and 159 synthetic samples were incorporated into the raw dataset, respectively.

## 2.2 Machine learning models

### 2.2.1 Building probability and classification machines

In supervised classification problems, ML algorithms utilize grouped attributes and features to identify patterns within human-curated datasets (Kalita, 2022). Samples in these datasets are typically assigned a label (class), target value, or dependent variable, which correspond to independent variables and features. The model utilizes this information to understand the relationships between the independent and dependent variables during the training process (Sendhilkumar and Geetha, 2023). The models are subsequently refined and evaluated for accuracy using a subset of the known classification dataset that has not been previously encountered by the model. A distinct validation set is employed to adjust the probability estimates. Numerous classification ML models employ probability estimates to determine the appropriate class (Murphy, 2012). When calibrated, these probability estimates can provide information that extends beyond merely identifying an individual's category but can also indicate the degree of likeness of an individual belonging to a category (Malley et al., 2012). Most ML algorithms, when initially deployed, lack calibration for precise probability predictions. Calibration is essential to ensure that the empirical probability is both valid and accurate (Dawid, 1985). In the absence of calibration, certain model outputs may push probability estimates toward 0 or 1, necessitating correction through calibration (Niculescu-Mizil and Caruana, 2005). Typically, either sigmoid ("Platt scaling") or isotonic regression is employed for calibration on a validation dataset that the model has not previously encountered (Niculescu-Mizil and Caruana, 2005). Subsequent to these steps, the model may be utilized to predict a class within a dataset where the classification remains unknown.

Five diverse algorithms were tested based on various methodological and practical reasons. First, we choose algorithms that could produce reliable probability estimates and have been widely utilized and validated (Malley et al., 2012; Wang et al., 2019). Algorithms were also chosen by performance on smaller tabular datasets, low computing resource requirements, and their availability in the scikit-learn Python library, which is available publicly for download (Pedregosa et al., 2011). These methods were chosen over more complex deep-learning methods, which often underperform on small tabular datasets (Grinsztajn et al., 2022) and require significant time and expertise for hyper-tuning (Mohammed and Kora, 2023); and other complex ensemble methods, which can require more computing resources without increased accuracy. Logistic regression (LR) is a parametric model akin to linear regression in its functionality, yet it is more appropriate for classification tasks (i.e., binary outcomes) (Hilbe, 2016). The analysis relies on the likelihood of an event occurring and the alignment of predictor response variables within the probability distribution (Hilbe, 2016). The algorithm is inherently calibrated for precise probability outputs due its foundational reliance on probability.

$K$-nearest neighbor (KNN), support vector machine (SVM), classification and regression tree (CART), and random forest (RF) are non-parametric models demonstrated to be effective in probability estimation following calibration (i.e., Niculescu-Mizil and Caruana, 2005; Kruppa et al., 2014; Dankowski and Ziegler, 2016; Cearns et al., 2020). KNN functions as a "lazy learner" by determining the distance between data points according to the characteristics of the training dataset (Geetha and Sendhilkumar, 2023). The "$K$" in KNN refers to a small positive integer that determines the number of neighbors taken into account when predicting the category of a data point (Geetha and Sendhilkumar, 2023). KNN is extensively employed in palaeosciences for paleoclimate regression issues, particularly through the modern analog technique (Simpson, 2007). The support vector machine (SVM) is a model that positions data items in $n$-dimensional space based on $n$ features (Geetha and Sendhilkumar, 2023). Classification is achieved by identifying a hyperplane in the dimensional space that distinguishes between the classes (Geetha and Sendhilkumar, 2023).

CART and RF are tree-based learning algorithms. Trees are formed through three fundamental steps: (1) binary splits are selected, (2) a determination is made about whether the node is terminal or requires further splitting, and (3) a class is assigned to the terminal leaf node (Bell, 1999). RF is founded on the principles of natural variability and randomness inherent in trees, where both the variables and the individual elements exhibit a degree of randomness (Genuer and Poggi, 2020). RF classification problems utilize a committee of decision trees that collectively vote to determine the predicted class (Hastie et al., 2009). In the classification problems, each vote corresponds to a classification in the terminal node of the tree (Malley et al., 2012), with the majority vote determining the final classification outcome. The probability estimates are derived by calculating the fraction of votes from each tree to determine the predicted class probability.

### 2.2.2 Verification, tuning, and calibration of models

The data were split into a $60 : 20 : 20$ training, testing, and validation set. This provided enough data to train the model with high accuracy and ensure that testing and calibration could occur on datasets that were previously unseen during training. The models underwent testing and hyperparameter tuning using a $k$-fold cross-validation approach, incorporating 10 data splits and a parameter grid with the test dataset. $K$-fold cross-validation involves partitioning the data into

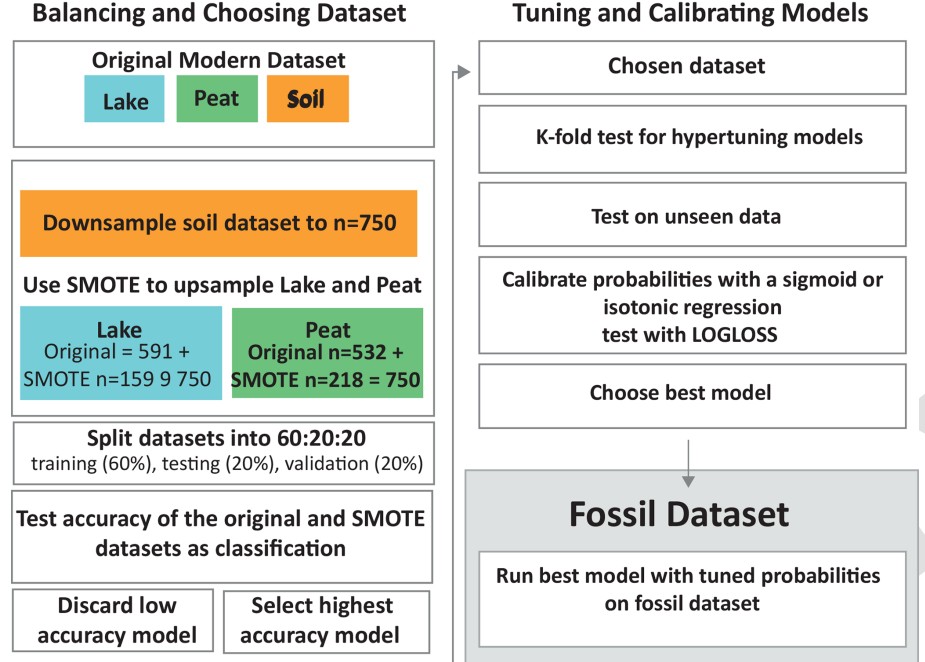

**Figure 2.** Illustration of the methods employed in this study for testing, tuning, and validating the datasets and models.

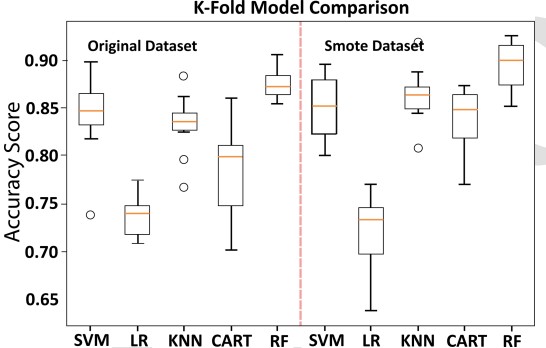

**Figure 3.** Comparison of $k$-fold testing models between datasets, utilizing $k$-fold cross-validation for classification on both SMOTE and original datasets. The $k$-fold comparison utilized 10 splits across supported vector machine (SVM), logistic regression (LR), $K$-nearest neighbor (KNN), classification and regression tree (CART), and random forest (RF).

equal-sized subsets, which are then utilized $k$ times, with $k-1$ subsets used for training and one subset reserved for validation (Jung, 2018). The performance is evaluated by averaging each $k$ iteration. The parameter grid facilitates the iteration over a finite set of values to identify optimal variables for tuning. After tuning, all models and datasets were retested for accuracy. The distribution was subsequently plotted, and the mean $F_1$ accuracy results were computed (Fig. 3).

### 2.2.3 Probability estimate calibration and application of classification machines

Instead of solely predicting the class (e.g., soil, peat, lake), we are employing the probability estimate output generated by the classification algorithms as a proxy for provenance change. The probability output enables the estimation of the likelihood that a specific sample belongs to a particular class, thus facilitating the identification of periods of mixed provenance. Due to the lack of calibration in the default probability estimates of the algorithms employed, we applied sigmoid and isotonic regression on the validation dataset to rectify any distortion and then assessed the effectiveness using the log-loss function in scikit-learn. Log loss is employed in probability scenarios where the likelihood of an event being true is represented as 1, equally true as 0.5, and false as 0 (Manzali et al., 2017). In log loss, a greater divergence between the predicted value and the actual value results in a higher log-loss score (Dembla, 2020). A lower score indicates greater accuracy in predictions. Log-loss scores were subsequently compared across models to evaluate performance. To estimate the 95 % confidence intervals for each downcore record, we performed 500 bootstrap resampling on the probability predictions. These were computed separately for each record to reflect their individual variance.

## 2.3 Application of models, downcore pollen, non-pollen palynomorph, XRF, and brGDGT analysis

To assess the accuracy of the probability estimates on the downcore record, five machine-learning models were applied to two published brGDGT records that included datasets of pollen and non-pollen palynomorphs (NPPs). Aquatic pollen and NPPs provide critical insights into alterations in lake or wetland ecology (e.g., Cromartie et al., 2020; Robles et al., 2022). We selected two records: one from Armenia in the southern Caucasus (Vanevan peat: 40°12′8.83″ N, 45°40′24.03″ E; Robles et al., 2022) and another from southern Spain (Padul paleolake: 37°00′39″ N, 3°36′14″ W; Camuera et al., 2018, 2019; Ramos-Román et al., 2018; Rodrigo-Gámiz et al., 2022), both situated at comparable latitudes in Eurasia.

The extraction methods for brGDGTs are detailed in the original articles by Robles et al. (2022) and Rodrigo-Gámiz et al. (2022). We revisited the original chromatograms of Robles et al. (2022) to investigate the presence of the IIIa″ isomer, which had not been published previously, to verify the brGDGT-based ML lake probability output. The IIIa″ isomer was reported in Rodrigo-Gámiz et al. (2022). Robles et al. (2022) provide identification and counting methods for aquatic pollen and NPPs in the Vanevan peat, while Ramos-Román et al. (2018) and Camuera et al. (2019) address similar methods for the Padul paleolake. Additionally, we re-calculated the reconstructed water depth based on aquatic pollen and NPPs. The analysis relies on the raw datasets employing the original equations established by Robles et al. (2022) and Camuera et al. (2019). Instead of applying a smoothing technique to the water-depth reconstruction, as done by Camuera et al. (2019) on the original 200 000-year-old sequence, we retained the original sample-to-sample curve for clarity to compare the shorter brGDGT sequence. These fossils of semi-aquatic plants and fungal and fern spores are generally representative of local change, particularly in wetlands (Gill et al., 2009; Tunno and Mensing, 2017), which strengthens their usage as local indicators. The percentages of the aquatic and NPP taxa were calculated by summing up all relevant pollen types for each record and dividing each taxon by the total sum. We calculated and re-calculated key brGDGT-based indices (Table 1) to compare our ML results with the brGDGT record as well as the aquatic pollen and NPPs. In addition, we also compared our results with the principal component analysis on the XRF datasets, also taken from the same cores, published in Robles et al. (2022) and Camuera et al. (2018). The descriptions of this analysis can be found in the original publications.

## 2.4 Descriptive statistics

The programming languages R (R core team, 2020) and Python (Python Software Foundation, Python Language Reference, version 3.7.3, available at http://www.python.org, last access: 24 April 2024) were utilized alongside ggplot2 (Wickham, 2016) and matplotlib (Hunter, 2007) to visualize the results. Redundancy analysis (RDA) was performed using the vegan package (Oksanen et al., 2019). RDA was employed in two capacities: (i) to compare the fractional abundances of the brGDGTs in the global modern dataset with the authors' descriptive categories (i.e., soil, peat, lake) and (ii) to compare the probability estimate results from the Vanevan and Padul records with the pollen and NPPs. In this analysis, we downsampled the pollen record to align with the brGDGT resolution, selecting samples that were no more than 100 years apart. Bayesian change-point analyses were conducted on the brGDGT-based ML lake probability results using the bcp package (Erdman and Emerson, 2007) in R to identify significant shifts in depositional environments.

## 3 Results

### 3.1 New modern brGDGT dataset

The raw database compiled for this study comprised a total of 2282 TS2 samples (591 from lakes, 532 from peats, and 1177 TS3 from soils). This addition includes 319 lake samples to the database of Martínez-Sosa et al. (2021), 62 peat samples to Naafs et al. (2017b), and 450 soil samples to the Dearing Crampton-Flood et al. (2020) datasets. Subsequent to the compilation of this dataset, additional datasets have been published (e.g., Raberg et al., 2022b; Martínez-Sosa et al., 2023) that are not incorporated into our dataset. Figures 1 and S1 in the Supplement illustrate the distribution of the brGDGT datasets.

### 3.2 Model accuracy and log loss

In classification mode, all models demonstrated mean accuracy $F_1$ scores, which measures the predictive accuracy of the models between 0.72 and 0.90 (Fig. 3). The study compared the performance of various classification models, with the SMOTE dataset showing superior results over the raw unbalanced dataset for SVM, KNN, CART, and RF (Table 2). The raw dataset showed better performance with LR, while the SMOTE dataset improved probability estimates for SVM and RF but decreased for LR, KNN, and CART. The sigmoid calibration improved probabilities for RF, CART, and KNN but decreased for SVM and LR. The isotonic calibration improved probabilities for KNN and CART but decreased for SVM, LR, and RF over uncalibrated probabilities. The sigmoid function outperformed the isotonic function on both datasets for SVM, LR, and RF. The RF model with the SMOTE dataset had the highest accuracy and the lowest log-loss score for sigmoid and uncalibrated probabilities and was therefore chosen for our downcore analysis as the best-performing model.

**Table 1.** The brGDGT indices employed in this study.

| Index | Formulae | Citation |
|---|---|---|
| $MBT'_{5ME}$ | $MBT'_{5ME} = ([Ia] + [Ib] + [Ic]) / ([Ia] + [Ib] + [Ic] + [IIa] + [IIb] + [IIc] + [IIIa])$ | De Jonge et al. (2014b) |
| $CBT'$ | $CBT = {}^{10}\log([Ic] + [IIa'] + [IIb'] + [IIc'] + [IIIa'] + [IIIb'] + [IIIc']) / ([Ia] + [IIa] + [IIIa])$ | De Jonge et al. (2014b) |

**Table 2.** Evaluation of accuracy across various models to determine optimal performance. The mean accuracy results were derived from a *k*-fold evaluation of the models, focusing on the classification of data categories (i.e., lake, peat, soil) using 10 splits. Log loss was computed for the probability estimates following their calibration, using either a sigmoid or an isotonic function. For these functions, values approaching 0 signify improved performance. Bold values represent the best-performing models once calibrated.

| Model and database | $F_1$ mean accuracy | Standard deviation | Log loss uncalibrated | Log loss sigmoid | Log loss isotonic |
|---|---|---|---|---|---|
| SVM | 0.84 | 0.04 | 0.46 | 0.51 | 0.93 |
| SVM_SMOTE | 0.85 | 0.03 | 0.40 | 0.50 | 0.64 |
| LR | 0.74 | 0.02 | 0.66 | 0.68 | 0.75 |
| LR_SMOTE | 0.72 | 0.04 | 0.69 | 0.70 | 0.73 |
| KNN | 0.83 | 0.03 | 1.17 | 0.46 | 0.80 |
| KNN_SMOTE | 0.86 | 0.03 | 2.08 | 0.41 | 0.41 |
| CART | 0.79 | 0.05 | 0.91 | 0.59 | 0.64 |
| CART_SMOTE | 0.84 | 0.03 | 3.95 | 0.55 | 0.55 |
| RF | 0.88 | 0.01 | 0.35 | **0.34** | 0.48 |
| RF_SMOTE | **0.89** | 0.03 | **0.31** | **0.3** | 0.56 |

## 3.3 Downcore analysis

### 3.3.1 Downcore probability estimates and change-point analysis

The Padul record showed mean probabilities for lake, peat, and soil, with lake having the highest probability in 68 out of 93 samples and peat in 25 out of 93 (Fig. 4, column 1). Vanevan's mean probability was 0.87 for lake, 0.05 for soil, and 0.08 for peat, with lake samples having the highest probability in 44 out of 46 samples, peat in 2 out of 46, and no samples having the highest probability in soil (Fig. 4, column 2).

### 3.3.2 Probability analysis by change-point phases

For the Padul record, change-points were detected at 12 837, 20 627, and 29617 cal BP for lake probabilities, dividing it into phases 1–4 (Fig. 4, column 1). Change-points in the Vanevan record were identified at 2043, 3577, 4628, 5061, and 8592 cal BP, based on lake probabilities (Fig. 4, column 2). Change-point analysis of the Padul record indicates that brGDGT-based ML lake probabilities peak in phase 3, while in phases 1, 2, and 4, these probabilities vary between soil and peat. BrGDGT-based ML soil probabilities are the highest in phase 1, while peat probabilities are consistent, primarily in phases 4 and 2 (Fig. 4, column 2). The probabilities of Vanevan brGDGT-based ML lake probabilities are elevated across the entire record, peaking during phases 5, 6, and 2, while peat probabilities are elevated in phases 4 and 1,

and soil probabilities exhibit fluctuations with peat and lake, predominantly in phases 4 and 3 (Fig. 4, column 2).

## 3.4 RDA analysis of modern and downcore pollen, NPP, and brGDGTs

### 3.4.1 Modern samples

The RDA analysis reveals the association of brGDGTs with each depositional unit (i.e., soil, lake, peat) in the global modern database and the downcore probability predictions. Most variance can be explained across RDA-1 (30.3), where peat and soil sources sit in contrast to lakes in the modern database (Fig. 5a). BrGDGTs Ia and IIa are more clearly associated with peat and soil depositional environments, while the rest have a stronger association with lake and soil environments (Fig. 5a). Comparisons between depositional environment probability estimates, aquatic pollen, and NPPs reveal relationships between these variables in the downcore record. Pollen and NPP associations between peat and lake probabilities include Cyperaceae pollen, while algae such as *Pediastrum*, *Botryococcus*, and *Myriophyllum* have associations with lake probabilities. For soil, spores and algae are associated with these depositional environments. Hdv-200 and Cyperaceae have the highest explanatory power for peat; monolete spores and *Polypodium* for soil; and *Botryococcus*, *Myriophyllum*, and *Pediastrum* for lakes (Fig. 5b and c).

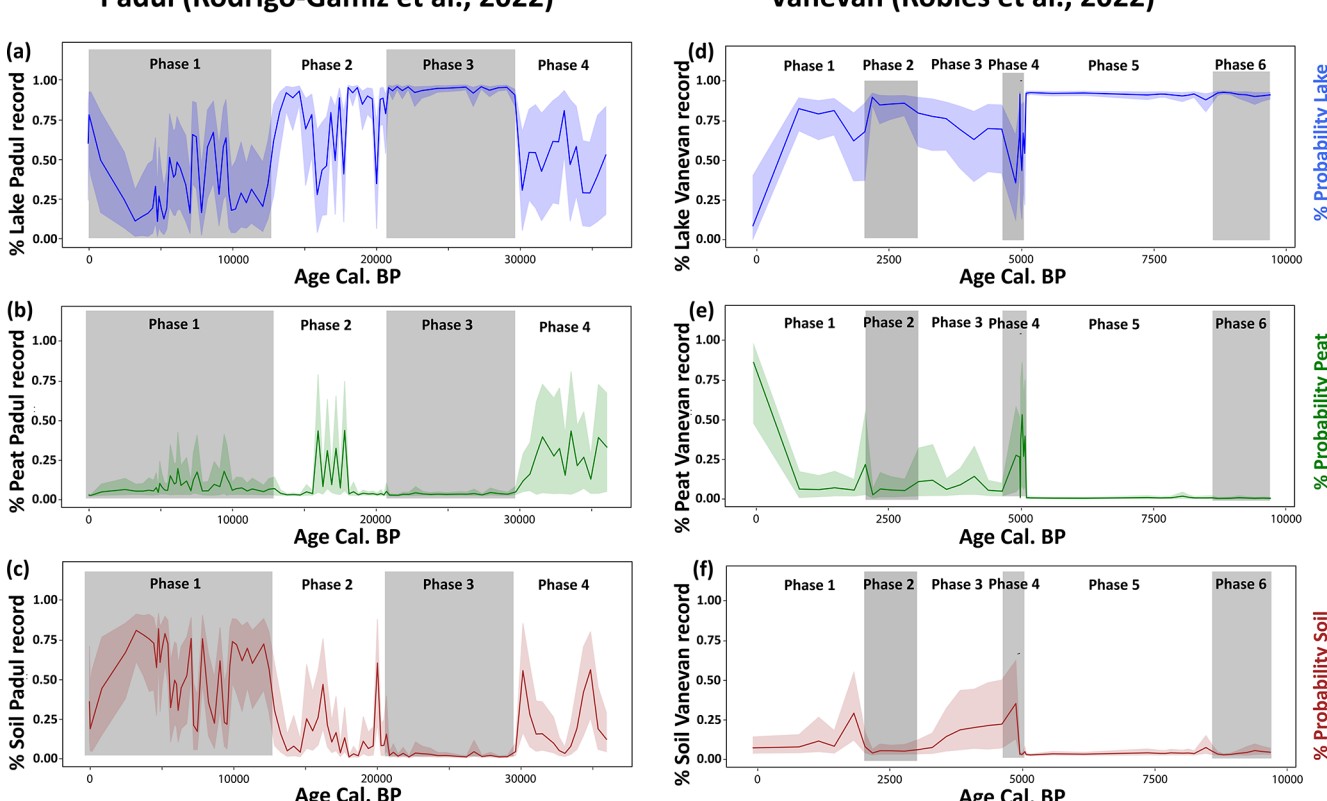

**Figure 4.** Downcore probability estimates with 95 % confidence intervals and change-point breaks from random forest (RF) on the SMOTE dataset with a sigmoid calibration. Results from the Padul (1) and Vanevan (2) records are broken down by lake probabilities (blue curves – **a**), peat probabilities (green curves – **b**), and soil probabilities (brown curves – **c**). Highlighted gray and white boxes indicate change-point mean breaks, identifying phases. Probability estimates from other models can be found in the Supplement (Figs. S9–S11).

## 4 Discussion

### 4.1 Probability estimates for chosen models and application to downcore records

#### 4.1.1 Model accuracy

The $F_1$ score evaluates the accuracy of a model's predictions of both precision (how many predicted positives were positive) and recall (from all the positives, how many positives the model predicted) and can balance between understanding false positives and false negatives (Boozary et al., 2025). This score allows for a more robust accuracy when measuring each model. Many things may explain differences in $F_1$ scores across our models. For example, KNN, SVM, and CART models are prone to overfitting (Huang et al., 2005; Berk, 2008; Jadhav and Channe, 2016), which may have accounted for their lower $F_1$ scores (Table 1). RF generally does not overfit due to its ability to handle noise in the datasets (Parmar et al., 2019), which may result in a higher $F_1$ score. While LR does not typically overfit, the lower $F_1$ score may be due to its assumptions of linearity (Nick and Campbell 2007), which may be problematic if

there is no clear division in the dataset. The balanced versus unbalanced datasets may have also impacted performance. RF generally handles unbalanced datasets well (Anaissi et al., 2013), and the SMOTE dataset only offered marginal improvements to the $F_1$ score, while CART's $F_1$ score was significantly improved with the balanced SMOTE datasets. For the log-loss scores, logistic regression is already calibrated (Kull et al., 2017) so calibrating may result in a lower log-loss score, with both sigmoid and isotonic calibration. SVM does not produce true probabilities by default and needs to be calibrated for these results (Kull et al., 2017). By calibrating them with a sigmoid or isotonic regression, the output turns to true probabilities which may result in a lower log-loss score. For KNN, CART, and RF, calibration improved the models on both datasets.

The random forest model utilizing the SMOTE dataset achieves an $F_1$ score of 89 % (Table 2) in classification, which is lower than the 95 % $F_1$ score reported for the Big-Mac model by Martínez-Sosa et al. (2023). The difference in $F_1$ scores is anticipated as a result of differing training datasets and methodologies, including the incorporation of isoprenoid GDGTs by Martínez-Sosa et al. (2023), their es-

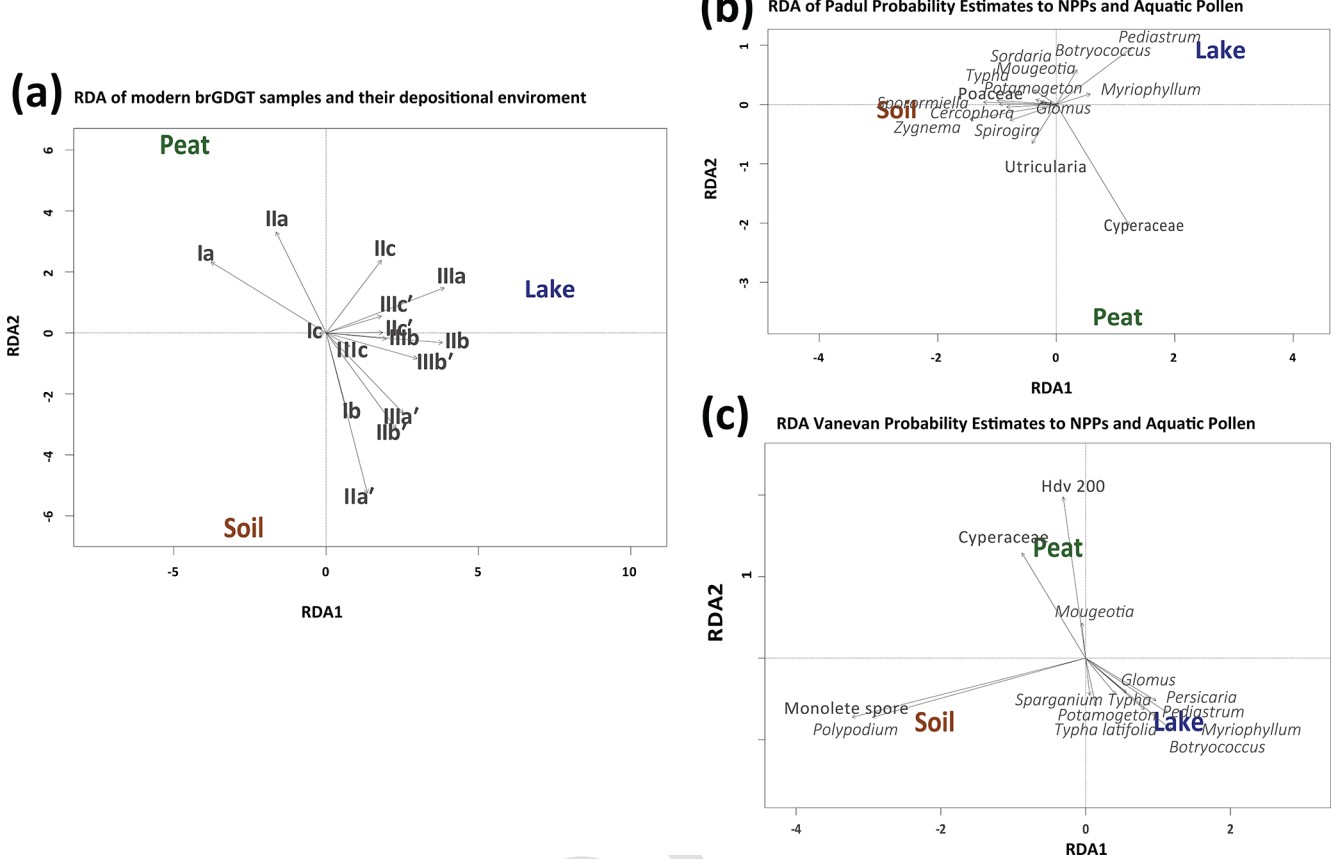

**Figure 5. (a)** RDA analysis of the modern fractional abundances of the global brGDGT database with their depositional environments. **(b)** Probability estimates results of the depositional environment compared with pollen and NPPs of the Padul record (i.e., Rodrigo-Gámiz et al., 2022; Camuera et al., 2019) and **(c)** the Vanevan record (i.e., Robles et al., 2022).

tablishment of curated clusters, and the application of classification models. Our models focus on probability estimate outputs instead of discrete classes, allowing for a nuanced understanding of shifts in provenance; thus, a lower $F_1$ score is permissible. A score of 89 % demonstrated a strong and precise model, despite being lower than expected. The difference between the new model and the BigMac model is also seen when applied to the downcore records: the BigMac model failed to predict soil classification for both cores, whereas our probabilities for soil were elevated in the Padul record (Figs. S6 and S7). The inclusion of additional soil samples in our database enhances soil identification during model training (BigMac $n = 192$, database in this paper $n = 750$).

## 4.1.2 Validating models with pollen, NPPs, XRF, and derived water-depth reconstructions

We evaluated the accuracy of our ML model for detecting provenance change by comparing the probability estimates it produced with published data on pollen, NPPs, XRF, and water-depth estimates derived from these proxies to check for quantitative similarities (Figs. 5 and 6). The comparison of GDGT-model variables with depositional environments indicates clear associations. Individual RDA analysis of both records associates Cyperaceae pollen, prevalent in wetland and peat contexts, with modern brGDGT samples obtained from depositional peat environments (Fig. 5). Downcore records indicate distinct associations between pollen and algae, specifically *Pediastrum* and *Botryococcus*, which are typically associated with open lakes, and the brGDGT-based ML probability estimates associated with the depositional lake environment (Fig. 5). Monolete spores in the Vanevan record, along with *Sordaria* and *Sporormiella* in the Padul record, are related to the brGDGT-trained ML probability estimates of depositional soil environments. In the Padul record, these spores were likely introduced through human activity (Ramos-Román et al., 2018) and may be associated with erosion into the lake. Pollen from semi-aquatic plants, including Cyperaceae and *Typha*, follow similar trends that align with brGDGT-trained ML lake probabilities, as well as increases in brGDGT-trained ML probabilities for peat and soil across both records (Figs. 6 and 7). The comparison of reconstructed water-depth results with the probabil-

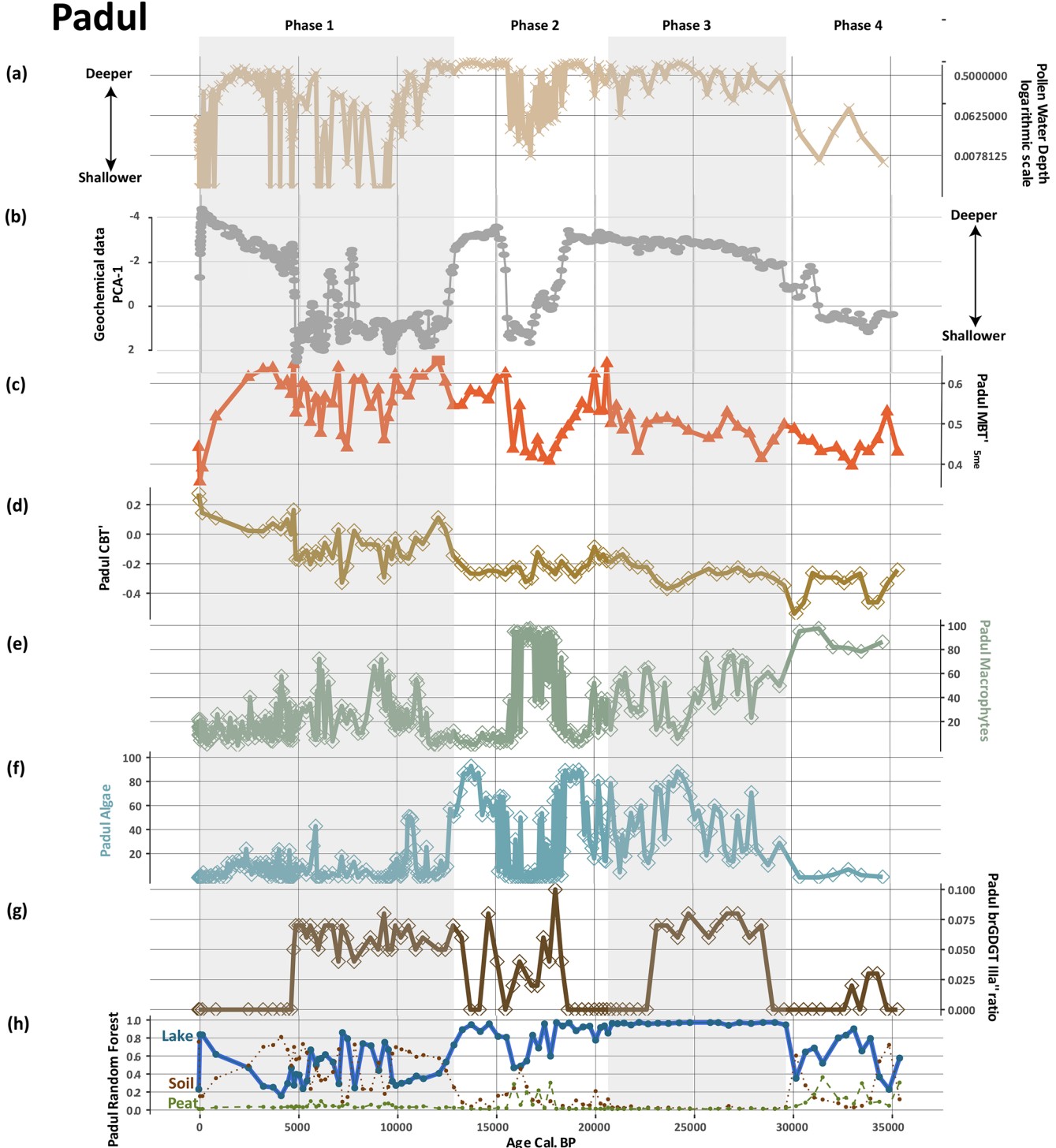

**Figure 6.** Comparison of probability estimates from the Padul record with aquatic pollen, NPPs, XRF, and brGDGT indexes (data from Ramos-Román et al., 2018; Camuera et al., 2018, 2019; Rodrigo-Gámiz et al., 2022). **(a)** Pollen- and NPP-based water-depth reconstructions. **(b)** Output from principal component analysis (PCA-1) from the XRF and geochemical data (Camuera et al., 2018). **(c)** The $MBT'_{5ME}$ brGDGT index. **(d)** The $CBT'$ brGDGT index. **(e)** Selected aquatic plants including Cyperaceae and *Typha*. **(f)** Selected algae *Pediastrum*, *Botryococcus*, and *Mougeota* as well as aquatic plants Cyperaceae and *Typha*. **(g)** The $IIIa''$ brGDGT ratio. **(h)** Probability estimates for the depositional lake environments (this study).

ity estimates from the brGDGT-trained ML model for both cores indicates that the two proxies exhibit similar quantitative patterns of increase and decrease. Similarly, PCA data derived from XRF datasets, from the same core, follow these trends. Robles et al. (2022) associate higher lake levels with the PCA-1 associated with positive loadings (P, K, Al, Mg, Si, Ti, Fe) and negative loadings (S) with lower lake levels. This suggests that our models accurately identify changes in sourcing and hydrology across both records (Fig. 7a). Robles et al. (2022) interpreted the water-depth changes for the Vanevan record based on aquatic pollen, NPPs, and XRF data identifying a shallow lake from 9700 to 9400 cal BP, a lake system from 9700 to 5100 cal BP, a transitional phase from 5100 to 4950 cal BP, and peatland development from 5100 cal BP to today. This aligns closely with our change-point phases, indicating elevated lake probabilities during phases 6 and 5, high peat probabilities during phase 4, and a rise in soil and peat probabilities from our brGDGT-trained ML model over the past 5000 years (Fig. 7e).

Principal component analysis (PCA) was also conducted on the XRF dataset on the Padul core and the authors used this as a proxy for lake-level change. Camuera et al. (2018) attributed negative loadings of PCA-1 with higher lake levels (Ca, Sr, Si, A, MS) and positive loadings with lower lake levels (Fe, S, Br, TOC, C/N). The ML-brGDGT-based probability estimates, the PCA output from geochemical data including XRF, and the pollen-based water-depth reconstructions are in alignment with Padul (Fig. 6), indicating the model's accuracy. The probability estimates in the Padul record exhibit trends analogous to the Vanevan results, with an alignment with water depth, as indicated by pollen and NPPs (Fig. 7b). The estimates derive from the pollen data and XRF data (Camuera et al., 2018, 2019), indicating a low water stand in phase 4, a high water stand in phase 3, a fluctuating high to low to high stand in phase 2, and a high fluctuating to low to high stand in phase 1. The observed trends are reflected in our brGDGT-based ML lake probability estimates. Similar to the Vanevan record, the Padul record predominantly features samples with brGDGT-based ML lake probabilities assigned to lakes. However, there is greater variation among categorical types. This is evident in phases 4 and 3 (where peat and soil probabilities are combined with lake probabilities) and in phase 1 (where notable fluctuations occur in soil and lake probabilities).

## 4.2 Environmental controls and depositional shifts in downcore brGDGT records

### 4.2.1 Identifying provenance changes in downcore records (and their impact on $\mathrm{MBT}'_{5ME}$)

The ML-probability estimates may be interpreted as originating from either a dominant or mixed-sourced sedimentary environment. During periods of high brGDGT-based ML lake probabilities, the findings indicate that brGDGTs are pro-

duced in situ in shallow lakes and wetlands, alongside contributions from other sources. This result is unexpected for the Vanevan record, as the brGDGTs from the last 5000 years exhibit a closer alignment with soil samples when plotted on a ternary diagram (Robles et al., 2022). We compared our results with the classification provided by the BigMac ML model from Martínez-Sosa et al. (2023), which classified most samples as a depositional lake environment, similar to our results, while the samples at 5000 cal BP for Vanevan were categorized as soil and peat rather than soil and lake. In the Padul record between 35 000 and 30 000 cal BP and from 10 000 cal BP to the present, differences between models include samples classified as soil (our model) rather than peat or lake (Fig. S3). A potential bias in both models may arise from samples in the global database categorized as lakes but with substantial contributions of brGDGTs from depositional soil or peat environments.

Second, our findings indicate that the identification of lacustrine brGDGTs produced in situ from depositional lake environments using a model provides more nuance than the quantification of the IIIa″ brGDGT isomers. Rodrigo-Gámiz et al. (2022) identified IIIa″ in the Padul record, which is attributed to in situ brGDGT lake production. Here, the ratio of brGDGTs IIIa″ aligns with the brGDGT-based ML lake-probability estimates for this record (Fig. 6). The IIIa″ isomer is completely absent from the Vanevan record; however, the brGDGT-based ML lake probability estimates approach 100 %. This supports earlier studies indicating that the IIIa″ isomer is not universally found in all lake systems (i.e., Weber et al., 2015; Dang et al., 2018). However, the lack of discussion regarding the absence of the IIIa″ isomer in both modern and downcore records is notable and warrants attention in future investigations.

The probability and RDA results underscore the need for multivariate methods in the analyses of depositional environments. RDA analysis of the global brGDGT database reveals a distinct separation along RDA-2 between 5- and 6- methyl pentamethylated brGDGTs in various modern depositional environments (Fig. 5a). This is observed between IIa and IIa′ associated with depositional peat and soil environments, respectively (Fig. 6). Martínez-Sosa et al. (2023) identified IIa′ as the most significant isomer for provenance classification using their random forest model, a finding that aligns with our models. Furthermore, brGDGT Ia exhibits a stronger association with depositional peat environments compared to other tetramethylated brGDGTs linked to lake environments, highlighting the need to advance beyond ternary diagrams for provenance identification.

The results of our models indicate that variations in brGDGT provenance, even in mixed sedimentary environments, significantly influence widely used indexes like $\mathrm{MBT}'_{5ME}$ (Figs. 6 and 7). Pollen and NPPs provide independent confirmation of the impact that these changes have on $\mathrm{MBT}'_{5ME}$, particularly when analyzed alongside data from our new brGDGTs global database. In this database,

# Vanevan

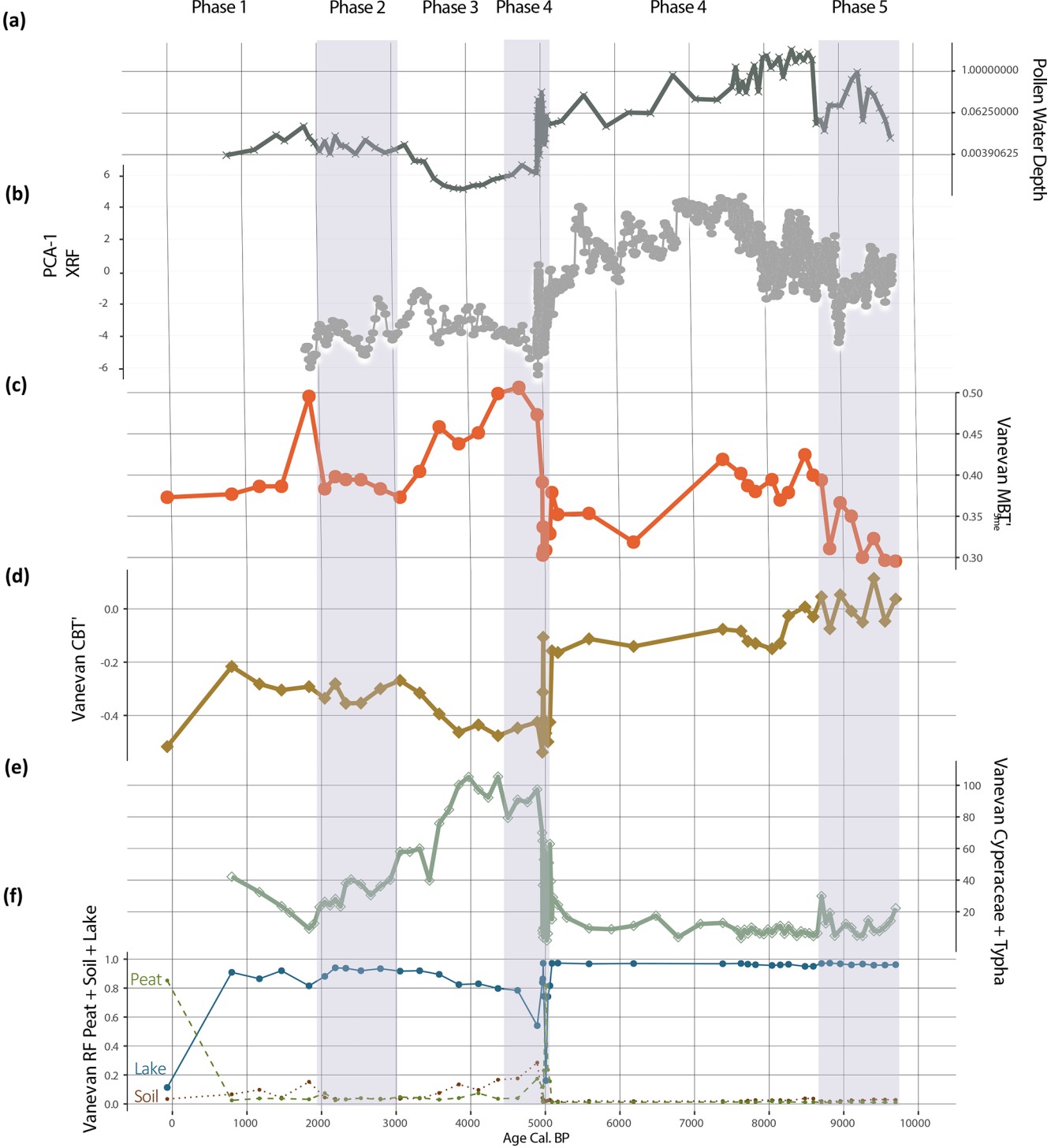

**Figure 7.** Comparison between the probability estimates on the Vanevan record and the aquatic pollen, NPPs, XRF, and brGDGT indexes (data from Robles et al., 2022). **(a)** Pollen- and NPP-based water-depth reconstructions. **(b)** PCA output on XRF datasets. **(c)** The MBT$'_{5ME}$ brGDGT index. **(d)** The CBT$'$ brGDGT index. **(e)** The selected algae and aquatic plants for the Vanevan record are *Pediastrum*, *Botryococcus*, *Mougeota*, Cyperaceae, and *Typha*. **(f)** Probability estimates for the depositional lake environments on both records.

soil (0.56) and peat (0.58) exhibit higher mean $MBT'_{5ME}$ values compared to the lower values observed in lakes (0.39) (Fig. S5). Both the Vanevan and Padul records indicate that $MBT'_{5ME}$ values are elevated during periods characterized by high brGDGT-based ML soil and peat probabilities, while they are reduced during periods of high lake probabilities. The pollen-based water-depth reconstructions, serving as an independent proxy, exhibit trends analogous to both the $MBT'_{5ME}$ and brGDGT-based ML probability estimates. These changes are documented in additional proxies from these records, including XRF and sediment analysis (e.g., Robles et al., 2022; Camuera et al., 2018), highlighting the necessity of identifying the appropriate depositional contexts.

Our findings indicate that even minor changes in provenance can affect the $MBT'_{5ME}$ and cyclization of branched tetraether (CBT') indexes. Where increased brGDGT-based ML soil probabilities occur in the Vanevan and Padul records, they do not reach the threshold, indicative of a complete depositional environment shift, instead indicating mixed provenance (Figs. 6 and 7). The Vanevan record indicates that the large shifts in $MBT'_{5ME}$ and CBT' occur with increased inputs of soil and peat brGDGTs during phases 3 and 4. In the Padul record, variations in CBT' correlate with increases in brGDGT-based soil ML probabilities, particularly during phase 2.

The co-occurrence of aquatic pollen, NPPs, and $MBT'_{5ME}$ variations indicates that provenance, rather than temperature, drives these changes. Increases in $MBT'_{5ME}$ during phases 3 and 4 of the Vanevan record correspond to shifts in aquatic pollen, indicating a transition from lake to peatland, driven by a local catchment fire event (Leroyer et al., 2016; Robles et al., 2022). The observed changes are inconsistent with regional climate reconstructions (i.e., Joannin et al., 2014; Cromartie et al., 2020), confirming that provenance change is the primary driver of this alteration.

### 4.2.2 Environmental drivers of provenance changes

The impact of provenance changes on $MBT'_{5ME}$ highlights various factors that can alter the distribution of brGDGTs over time, making it a crucial aspect for environmental reconstructions. The environmental changes that cause a change in GDGT provenance will also affect the environmental chemistry. While large pH changes have the potential to impact $MBT'_{5ME}$ values in soils, muted pH changes in soils and the impact on GDGTs produced in lakes are less well constrained. The introduction of soil brGDGTs into a lake, even in small amounts, can alter the $MBT'_{5ME}$ distribution and also potentially introduce pH-related changes.

Rodrigo-Gámiz et al. (2022) identified a relationship in the Padul record between increases in reconstructed pH and MAAT variability within the upper 116 cm, approximately correlating to the last 5000 years. They associated this with a dried ephemeral lake and suggested potential bias in the

$MBT'_{5ME}$ reconstruction. In this section of the Padul record, high brGDGT-based ML soil probabilities align with increased CBT' and $MBT'_{5ME}$ values, indicating the contribution of soil-derived brGDGTs and potentially pH to this variation (Fig. 7). Soil provenance changes appear to exert a greater influence on $MBT'_{5ME}$ compared to peat and are more prevalent in the record; however, brGDGT-based ML probability estimates for these depositional environments overlap in certain sections of both records.

Our results also highlight the impact of both sudden and gradual depositional changes on the distribution of brGDGTs, driven by hydrological and ecological shifts. Hydrological changes, including variations in water depth in lakes (Stefanescu et al., 2021) and alterations in water-table levels in peat (Ofiti et al., 2024), have been demonstrated to affect brGDGT distribution. The water-depth equations for the Padul and Vanevan records incorporate Cyperaceae pollen at one end of the equation. Cyperaceae is typically associated with the development of wetland ecosystems and the process of lake shallowing. A correlation is observed between $MBT'_{5ME}$ and Cyperaceae for the Padul record ($-0.52$, $p$-value: $< 0.001$). There is a correlation between $MBT'_{5ME}$ and the water-depth reconstruction for the Vanevan record (0.40, $p$-value: 0.006). Alterations in hydrology influence shifts in ecological communities, which may or may not be driven by climate. Our results indicate that wetland development, resulting from ecological shifts such as the introduction of aquatic plants and/or lake shallowing, can influence $MBT'_{5ME}$ and the distribution of brGDGTs (Figs. 6 and 7).

### 4.2.3 Considerations for application to sedimentary sequences

The samples from the modern training dataset come from diverse modern environmental contexts, making our script applicable to most paleoenvironmental reconstructions. The probability outputs on the Padul core, which is 36 000 years old, align with the pollen and XRF water-depth reconstructions during the last glacial period (Fig. 6), confirming the model's usefulness for records extending beyond the Holocene. Distributions of samples across the Köppen–Geiger climate gradient are not balanced (Fig. S2), with temperate environments well represented, but tropical, arid, and arctic conditions are underrepresented. Considering this, caution is urged when applying this model in these environments. In addition, caution must be taken for records in deep time, where no current modern analogs exist.

The log-loss score of 0.31 for RF with a sigmoid calibration suggests that the downcore predicted probabilities accurately detect sediment change, even with mixed provenance. The 95 % confidence intervals on the downcore predictions, however, can vary throughout the sediment sequence, and care must be taken when applying the models to ensure the records' accuracy. Due to the nature of the modern database,

which relies on the correct sedimentary context identified by the original authors, some accuracy uncertainties are possible, even on a well-trained model.

## 4.3 Limitations and future directions

Our findings indicate that multivariate methods, such as machine learning, are needed for analyzing brGDGT distributions. To effectively utilize these tools, a standardized collection of datasets across research groups is essential, along with increased datasets from a variety of environments. A notable limitation of our study was our reliance on the published sample name. The identification of depositional environments was successful with these names; however, variations within a depositional environment, for example, shallow versus deep lakes, remain inadequately represented. To effectively utilize these tools, it is essential to collect additional information, including water depth, salinity, pH, and redox conditions, in a standardized manner across research teams.

This study demonstrates the necessity of multi-proxy approaches to comprehend the influence of ecological, hydrological, and depositional changes on brGDGT-based reconstructions. Numerous studies presently employ pollen-based climate reconstructions in conjunction with brGDGT reconstructions (e.g., Watson et al., 2018; Martin et al., 2019; Dugerdil et al., 2021c; Robles et al., 2022, 2023; Stefanescu et al., 2021). The findings indicate that aquatic pollen, NPPs, and XRF provide valuable insights for understanding biases introduced by alterations in depositional environments and provenance.

Many downcore studies are derived from smaller lakes, wetlands, and peatlands (e.g., Martin et al., 2019; Dugerdil et al., 2021c; Robles et al., 2022, 2023; Ramos-Román et al., 2022; Acharya et al., 2023; Barhoumi et al., 2024). This study emphasizes the importance of comprehending changes in depositional environments over geological time and advocates for studies in smaller lakes, wetlands, and peatlands. There is a particular necessity for the improved classification of wetland environments in the brGDGT literature.

## 5 Conclusion

This study demonstrated that multivariate methods enhance the understanding of how provenance changes affect brGDGT distributions and $MBT'_{5ME}$. A new database of modern samples ($n = 2301$ samples) has been utilized to apply probability estimates from five machine learning algorithms to downcore sediments, facilitating the identification of changes in brGDGT provenance across depositional lake, soil, and peat environments. Utilizing calibrated probability estimates enhances the identification of the provenance of brGDGTs, including those originating from mixed sources.

The results indicate that alterations in provenance, depositional environments, and hydrology, particularly the transition from open lakes to wetlands and variations in water depth, can substantially influence the brGDGT signal. The introduction of soil-derived brGDGTs, even in minimal quantities, significantly influences the brGDGT distribution and $MBT'_{5ME}$.

This study confirms that independent proxies, including aquatic pollen, non-pollen palynomorphs, and XRF, can effectively quantify hydrological and ecological changes, thereby influencing the gradual depositional alterations that may affect brGDGT distribution. Our models can accurately and independently identify changes in provenance and are applicable to existing global paleoenvironmental datasets. We suggest that complementary environmental proxies, including fossil pollen, non-pollen palynomorphs, XRF, diatoms, and testate amoebae, among others, are essential for confirming changes in provenance in brGDGT environmental reconstructions.

*Code availability.* The code and data for this project are publicly available at https://doi.org/10.5281/zenodo.17459703 (Cromartie, 2025b).

*Data availability.* Data for this paper are available at https://doi.org/10.17632/tr8tppy9fz.1 (Cromartie, 2025a).

*Supplement.* The supplement related to this article is available online at [the link will be implemented upon publication].

*Author contributions.* AC conceptualized the project. AC and CDJ performed the data curation. AC created the methodology. AC wrote the software and did the formal analysis. AC, CDJ, GM, LD, MR, MJRR, and SJ provided the validation. AC, SJ, GM, and CDJ provided the funding acquisition. AC, SJ, and GM did the project administration. MR, MRG, JC, MJRR, and GJM provided the resources. SJ, GM, LS, and OP provided the supervision. AC prepared the original draft. AC, CDJ, and GM wrote the subsequent drafts. AC, CDJ, GM, MR, LD, OP, MRG, JC, MJRR, GJM, CC, LS, and SJ did the reviewing and editing.

*Competing interests.* The contact author has declared that none of the authors has any competing interests.

ther geographical representation in this paper. While Copernicus Publications makes every effort to include appropriate place names, the final responsibility lies with the authors.

*Acknowledgements.* We would like to thank the reviewers for their thoughtful comments that improved this paper. We would like to thank members of the GDGT community who provided additional information in order to complete the GDGT database. ChatGPT was utilized to help streamline and improve performance for both the R and Python code utilized in this project. This is an ISEM contribution ISEM 2025-145.

*Financial support.* This research was funded by a National Science Foundation Graduate Research Fellowship (DGE-1650441) for A. Cromartie, a Chateaubriand "Make Our Planet Great Again" fellowship from the Embassy of France in the United States for A. Cromartie, and an iSite Muse mobility grant from the University of Montpellier for A. Cromartie. This research was funded, in whole or in part, by ANR, grant ANR-22-CE27-0018-02. A CC-BY public copyright license has been applied by the authors to the present document and will be applied to all subsequent versions up to the Author Accepted Manuscript arising from this submission, in accordance with the grant's open access conditions. CDJ received an SNSF PRIMA (grant no. 179783).

*Review statement.* This paper was edited by Petr Kuneš and reviewed by Joseph B. Novak and one anonymous referee.

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

**Remarks from the typesetter**

**TS3**    Prior to submitting the article we added an additional 20 samples to bring the soil sample count up to 1197 and the total db count up to 2301. Unfortunately, we did not catch that this section needed to be updated prior to submission and through peer-review. The correct number is on the methods section and the data in figure 2 is correct, and the total number of the db is correct in the conclusion and introduction. The data and information used in the article is based on the db number of 2301 and this change just reflects having an accurate number throughout the article.