# Peer review of "Utilizing Probability Estimates from Machine Learning and Pollen to Understand the Depositional Influences on Branched GDGT in Wetlands, Peatlands, and Lakes"

_EGUsphere, 2025_

## Referee Comment (RC1)

**Review of Cromartie *et al.* for *Biogeosciences***

**Joseph B. Novak**

**Recommendation**

Major revision.

**Summary**

Cromartie et al. present a new machine learning approach to probabilistically assess the provenance of brGDGTs in terrestrial sedimentary archives. This work improves upon previous work by Martinez-Sosa et al. (2023), the BIGMAC algorithm, by generating probability estimates that permit analysis of the likely relative contributions of brGDGTs from various sources to a sediment sample rather than discrete sample classifications. The improvement upon the BIGMAC algorithm is a contribution towards ongoing efforts to utilize brGDGTs as proxies of past climate change in the geologic record.

The writing is mostly clear, although there are some places where I was confused by the word choice or sentence structure. Wherever possible, I provided suggestions to revise the wording for clarity. I urge the editors to find a machine learning expert to evaluate the methodology of this work, as this technique does not fall within my expertise. My recommendation for a major revision is based upon my concerns regarding section 4.2.4 where a new brGDGT wetlands index is proposed (see major comments).

I look forward to the publication of this work after my comments are addressed.

**Major Comments**

*Introduction*

The introduction would benefit from some clarification as to why it was necessary to use five machine learning techniques to generate the model described here. Did you try five machine learning methods and then settle on one as the best? Are you somehow combining the output of all five models? Machine learning is generally a confusing (and intimidating!) methodology for many people, including some who would want to use your algorithm. Clarity on why you took this approach will make people more likely to understand what you did and therefore more likely to use your algorithm (and cite your work! 😊 ). I think an additional 1–3 sentences in the paragraph at lines 81–92 would be very helpful for clarifying this point.

*Materials and Methods*

L220–221: Do you mean that you are using the probability estimates as a means of understanding changes in brGDGT provenance through time? Because that is a different thing than using them as an environmental proxy. Please clarify.

*Discussion*

Figure 6a and 7a: why is the pollen water depth reconstruction plotted on a log scale? This seems a bit odd, should this not be plotted on a linear scale? Please explain.

Section 4.2.4: I question whether including this section distracts from the larger point of the paper.

Is this index not redundant since you are tracking basically the same thing with the % peat probability? Perhaps more importantly, there should be some sort of validation regarding whether this index is useful for identifying wetlands in a modern dataset. For example, is this index value higher in modern samples from wetlands than in those from dry soils or lakes? I think this section generally should be expanded upon significantly if the authors want to claim that this index can be used this way.

**Minor Comments**

*Abstract*

L21: I think you mean "Branched glycerol dialkyl glycerol tetraethers (brGDGTs) **are** critical molecular biomarkers" rather than "…serve as critical molecular biomarkers."

L22–23: Is the sentence starting with "Despite their success…" necessary?

L25: here and throughout, make sure to use the proper prime symbol ′

L25–26: "…where ecosystems are sensitive to diverse environmental factors." Do you mean that depositional environments in arid and semi-arid regions are prone to change in response to water stress?

 L31: "…obtained from the identical records." Do you mean from the same samples? Or from the same cores?

L34: Typo. "brGDGT provenance" not "brGDGTs provenance."

L36: I think a word is missing here. Do you mean potential biases in brGDGT paleotemperature reconstructions?

*Introduction*

L39–43: These two sentences are largely redundant. Could they be combined?

L47: "**their** potential" not "its potential." I would consider removing this initial clause in the sentence and starting with "A key challenge…" as this is a more focused start to the paragraph.

L47–48: I think you need a citation here since this thought is informed by previous work.

L56–58: I think you mean "The MBT′$_{5Me}$ index is correlated to temperature in lake sediments, peats, and soils [CITATIONS]."

L59–65: Somewhere in here it would be good to mention that MBT′$_{5Me}$ is systematically higher in soils than in lakes. This is usually the major source of concern when it comes to dealing with brGDGTs from potentially mixed sources, at least in lake sediments.

L78: what do you mean by ecological changes? As in bacterial ecology? This may be a word choice issue, I was really surprised to see the word "ecology" here.

L99 vs L100: do you mean depositional or provenance? Because those are different things. The words cannot be used interchangeably.

*Materials and Methods*

L123: I think you mean limited data, not limited publication.

L135: "A $C_{46}$ **internal** standard."

L139: "…the $C_{46}$ **internal** standard."

L176: Check the journal's referencing policies. I am not sure that "ibid" is permitted within this citation style.

L189–191: This sentence is missing a verb. Please clarify.

L192: same comment regarding ibid.

L196: same comment regarding ibid.

Figure 2: consider making the background color for the soil dataset symbol a lighter shade of brown. I found it hard to read the dark font against the dark brown background.

L222: do you mean "periods of mixed brGDGT provenance?"

L244–245: why did you retain the original sample to sample curve?

L252–254: I am not sure this is necessary to explain.

**Results**

L279–280: these sentences should be combined.

**Discussion**

L362: I think you mean that the two timeseries are qualitatively similar.

L402: italicize *in situ*

L429–431: I am not sure what you mean. The De Jonge et al. 2024 study found that MBT'$_{5Me}$ are generally reproducible between laboratories.

L450: the ' symbol should not be subscripted

L468: you can simply report this p-value as "p < 0.001"

L475–482: why are some of the brGDGTs written in square brackets sometimes but not other times?

---

## Author Response (AR1)

Dear Reviewer Novak,

Thank you for your thoughtful response on our manuscript. We have gone ahead and read through your reviews and answered them as best as we can. You will find our responses below. We hope that we sufficiently answered your comments and concerns below.

Review of Cromartie et al. for Biogeosciences

Joseph B. Novak

Recommendation

Major revision.

Summary

Cromartie et al. present a new machine learning approach to probabilistically assess the provenance of brGDGTs in terrestrial sedimentary archives. This work improves upon previous work by Martinez-Sosa et al. (2023), the BIGMAC algorithm, by generating probability estimates that permit analysis of the likely relative contributions of brGDGTs from various sources to a sediment sample rather than discrete sample classifications. The improvement upon the BIGMAC algorithm is a contribution towards ongoing efforts to utilize brGDGTs as proxies of past climate change in the geologic record.

The writing is mostly clear, although there are some places where I was confused by the word choice or sentence structure. Wherever possible, I provided suggestions to revise the wording for clarity. I urge the editors to find a machine learning expert to evaluate the methodology of this work, as this technique does not fall within my expertise. My recommendation for a major revision is based upon my concerns regarding section 4.2.4 where a new brGDGT wetlands index is proposed (see major comments).

I look forward to the publication of this work after my comments are addressed.

Major Comments

Introduction

The introduction would benefit from some clarification as to why it was necessary to use five machine learning techniques to generate the model described here. Did you

try five machine learning methods and then settle on one as the best? Are you somehow combining the output of all five models? Machine learning is generally a confusing (and intimidating!) methodology for many people, including some who would want to use your algorithm. Clarity on why you took this approach will make people more likely to understand what you did and therefore more likely to use your algorithm (and cite your work! )

- I think an additional 1-3 sentences in the paragraph at lines 81-92 would be very helpful for clarifying this point.

Response: Thank you for this response we have added a bit more information in the introduction on why these models were chosen.

Introduction: "We test five popular parametric and non-parametric machine learning models based on their ability to handle small tabular datasets and produce reliable probability estimates when calibrated (Malley et al., 2012; Wang et al., 2019). Models utilizing different structures were chosen, including simple tree-based algorithms (CART), ensemble trees (RF), linear models (LR), margin-based classifiers (SVM), and instance-based lazy learners (K-NN) to evaluate performance. The best-performing model was then chosen to apply to two down-core sedimentary sequences. "

Materials and Methods

L220–221: Do you mean that you are using the probability estimates as a means of understanding changes in brGDGT provenance through time? Because that is a different thing than using them as an environmental proxy. Please clarify.

Response: Yes, we are primarily looking at them to explain provenance change rather than as an environmental proxy in itself. We have gone ahead and removed environmental and added provenance. This sentence is now: "proxy for provenance change"

Discussion

Figure 6a and 7a: why is the pollen water depth reconstruction plotted on a log scale? This seems a bit odd, should this not be plotted on a linear scale? Please explain.

Response: We decided to plot the water depth reconstruction on a logarithmic scale following the precedent of the author's (i.e., Robles et al., 2022) original peer-reviewed articles rather than deviate. It is common to plot these pollen-based reconstructions in this way in order to visualize changes more clearly the results.

Section 4.2.4: I question whether including this section distracts from the larger point of the paper. Is this index not redundant since you are tracking basically the same thing with the % peat probability? Perhaps more importantly, there should be some sort of validation regarding whether this index is useful for identifying wetlands in a modern dataset. For example, is this index value higher in modern samples from wetlands than in those from dry soils or lakes? I think this section generally should be expanded upon significantly if the authors want to claim that this index can be used this way.

Response: After consideration, and since both reviewers mentioned the index as problematic, we have gone ahead and removed this index at this time since it will require major additions to the manuscript which distracts from the primary goal of utilizing machine learning. We will try to publish this index in the future with more information. To accommodate this change we have removed the discussion of this index from the abstract (lines 35-36). We removed figure 8 and section 4.2.4. This section has been changed to include a more through discussion of utilizing the brGDGTs in different environmental settings and confidence intervals for reviewer 2.

Minor Comments

Abstract

L21: I think you mean "Branched glycerol dialkyl glycerol tetraethers (brGDGTs) are critical molecular biomarkers" rather than "…serve as critical molecular biomarkers."

Response: We removed the "serve as" and changed to "are critical biomarkers"

L22–23: Is the sentence starting with "Despite their success…" necessary?

Response: We have removed the phrase "Despite their success, and instead start the sentence with "A key challenge"

L25: here and throughout, make sure to use the proper prime symbol '

Response: This has been updated throughout the manuscript. Thank you for catching this.

L25–26: "…where ecosystems are sensitive to diverse environmental factors." Do you mean that depositional environments in arid and semi-arid regions are prone to change in response to water stress?

Response: Yes, this is indeed what we mean, we have added additional text that says "where ecosystems are sensitive to diverse environmental factors including water stress from increased aridity"

L31: "…obtained from the identical records." Do you mean from the same samples? Or from the same cores?

Response: These samples are from the same cores. The texts has been updated to reflect that: "taken from the same sedimentary core sequence"

L34: Typo. "brGDGT provenance" not "brGDGTs provenance."

Response: Thank you this has been updated to correct this typo

L36: I think a word is missing here. Do you mean potential biases in brGDGT paleotemperature reconstructions?

Response: This sentence was removed due to remove the wetland index.

Introduction

Introduction

L39–43: These two sentences are largely redundant. Could they be combined?

Response: Thank you for this comment, although we can see where there are some redundancies, we have kept the sentences in the current form in order to make sure that the reader has the relevant literature.

L47: "their potential" not "its potential." I would consider removing this initial clause in the sentence and starting with "A key challenge…" as this is a more focused start to the paragraph.

Response: we have removed this phrase and start with "A key challenge" as you suggested.

L47–48: I think you need a citation here since this thought is informed by previous work.

Response : We have added the following citations to this line: (De Jonge et al., 2014 ; Naafs et al., 2017 ; Dearing Crampton-Flood, 2020; Martínez-Sosa et al., 2020; Raberg et al., 2022)

L56–58: I think you mean "The MBT'5Me index is correlated to temperature in lake sediments, peats, and soils [CITATIONS]."

Response: We have gone and updated this sentence to the following:

Text added:  The MBT'$_{5ME}$ index has been successfully utilized as grounds for various global temperature calibrations, because of its strong correlation to temperature in modern samples, concerning lakes, peats, and soils (e.g., De Jonge et al., 2014a; Hopmans et al., 2016; Naafs et al., 2017a; Dearing Crampton-Flood et al., 2020; Martínez-Sosa et al., 2021; Véquaud et al., 2022).

L59–65: Somewhere in here it would be good to mention that MBT'5Me is systematically higher in soils than in lakes. This is usually the major source of concern when it comes to dealing with brGDGTs from potentially mixed sources, at least in lake sediments.

Response: We agree we have updated the text with the following texts

Text added: "Provenance changes may introduce bias to temperature reconstructions based on the MBT'$_{5ME}$ index values generally being higher in soils than in lakes (Pablo Martínez-Sosa et al., 2021). "

L78: what do you mean by ecological changes? As in bacterial ecology? This may be a word choice issue, I was really surprised to see the word "ecology" here.

Response: This references the ecological changes within the wetlands, peatlands, and lake environments, not the bacterial ecology, however, we do understand this confusion so we have removed the word ecology from this sentence. The sentence now reads: "This paper presents a strategy for identifying provenance changes across"

L99 vs L100: do you mean depositional or provenance? Because those are different things. The words cannot be used interchangeable.

Response: We changed the wording to reflect that these changes in provenance by adding the following to the sentence "we demonstrate how these complementary proxies can aid in identifying potential hydrological, ecological, and depositional changes that may cause provenance shifts, introducing bias in brGDGT reconstruction"

Materials and Methods

L123: I think you mean limited data, not limited publication.

Response: removed publication and changed to "limited data"

L135: "A C46 internal standard."

Response: This has been updated

L139: "...the C46 internal standard."

Response: This has been updated.

L176: Check the journal's referencing policies. I am not sure that "ibid" is permitted within this citation style.

Response: We have changed to include the reference removing ibid.

L189–191: This sentence is missing a verb. Please clarify.

Response: I was unable to find a missing verb in these sentences, in this case "are" and "demonstrated" are the verbs and "functions" is the verb in the next sentence.

L192: same comment regarding ibid.

Response: updated

L196: same comment regarding ibid.

Response: updated

Figure 2: consider making the background color for the soil dataset symbol a lighter shade of brown. I found it hard to read the dark font against the dark brown background.

Response: We have gone ahead and updated this figure to a lighter shade of brown.

L222: do you mean "periods of mixed brGDGT provenance?"

Response: Yes, this refers to mixed provenance. We have gone ahead and removed the word sourcing and change it to provenance "thus facilitating the identification of periods of mixed provenance."

L244–245: why did you retain the original sample to sample curve?

Response: We retained the original sample to sample curve, because the original publication of the pollen record and water depth reconstructions was 200,000 years old making it difficult to see variations within the shorter (36,000 years) brGDGT record. We added additional text to clarify:

"Instead of applying a smoothing technique to the water-depth reconstruction, as done by Camuera et al., (2019) on the original 200,000-year-old sequence, we retained the original sample-to-sample curve for clarity to compare to the shorter brGDGT sequence."

L252–254: I am not sure this is necessary to explain.

Response: Although we agree we do not necessarily need to explain the plots, we have decided to keep the reference and description in the manuscript for clarity and to allow for the proper citations to the programs used for our analysis.

**Results**

L279–280: these sentences should be combined.

Response: We have gone ahead and combined these sentences as the following "The RF model with the SMOTE dataset had the highest accuracy and the lowest Log loss score for sigmoid calibrated probabilities and therefore was chosen for our analysis as the best performing model."

**Discussion**

L362: I think you mean that the two timeseries are qualitatively similar.

Response: We have added the following to the test reflect this change

"We evaluated the accuracy of our ML model for detecting provenance change by comparing the probability estimates it produced with published data on pollen, NPPs, XRF and water depth estimates derived from these proxies and to check the that the results for quantitative similarities (Fig. 5 and 6). "

L402: italicize in situ

Response: this has been updated

L429–431: I am not sure what you mean. The De Jonge et al. 2024 study found that MBT'

5Me are generally reproducible between laboratories.

Response: Thank you for this catch, indeed the sentence is indeed incorrect and phrased strangely. We have removed this sentence from the manuscript.

450: the ' symbol should not be subscripted

Response: This has been updated

L468: you can simply report this p-value as "p < 0.001"

Response: This has been changed to "p < 0.001"

L475–482: why are some of the brGDGTs written in square brackets sometimes but not other times?

Response: We have gone ahead and updated the manuscript to included the square brackets around all the brGDGTs when appropriate
**Citation**: https://doi.org/10.5194/egusphere-2025-526-AC1

Dear Reviewer 2,

We thank the reviewer for their time and suggestions which strengthen this manuscript. Below you will find our response to review #2 and the relevant changes made in the text. We hope we sufficiently responded to your thoughtful advice and suggestions.

The manuscript presents an interesting strategy on using machine learning probability estimates and pollen data to understand how source changes affect the distribution of brGDGTs and particularly the MBT'5ME index in sedimentary records. The study applies probability estimates from machine learning models to an extended modern global brGDGT database, to detect brGDGT contributions from different sources (soil, peat, lake) over time in two sediment archives and validate these findings through comparisons with independent pollen ad NPP data.

Although the study builds on previous research (Martinez-Sosa et al, 2023), it introduces several novel additions, including: addition of new modern br-GDGT samples to the training dataset, exploration of different probability calibration techniques and proposing a new "brGDGT wetland index". This index, if validated further, may help distinguish wetland-influenced brGDGTs from other depositional sources, with potential applications beyond brGDGT provenance analysis.

However, I find some aspects unclear or insufficiently explained:

1) The study uses five different machine learning models, but the criteria for selecting these specific models are not well justified. Why were these models chosen over others, such as deep learning approaches or ensemble methods beyond Random Forest? Also, the manuscript could benefit from a discussion of why certain models performed better and why others underperformed.

Response: In order to simplify the manuscript an ensure that the brGDGTs were the main focus we did not present an in-depth discussion of the selection of the ML models. However, early screening included utilizing other ensemble methods like XGBoost which performed similarly to Random forests but was computationally heavy and neural networks, which did not perform well. In the end we also stayed away from deep-learning models because they generally do not perform well on tabular datasets. We agree that some additional information is needed on model performance as well as model choice so we added the following text.

Introduction: "We test five popular parametric and non-parametric machine learning models based on their ability to handle small tabular datasets and produce reliable probability estimates when calibrated (Malley et al., 2012; Wang et al., 2019). Models utilizing different structures were chosen, including simple tree-based algorithms (CART), ensemble trees (RF), linear models (LR), margin-based classifiers (SVM), and instance-based lazy learners (K-NN) to evaluate performance. The best-performing model was then chosen to apply to two down-core sedimentary sequences."

Section added Methods:

"Five diverse algorithms were tested based on various methodological and practical reasons. Firstly, we choose algorithms that could produce reliable probability estimates and have been widely utilized and validated (Malley et al., 2012; Wang et al., 2019). Algorithms were also chosen by performance on smaller tabular datasets, low computing resource requirements, and their availability in the Scikit Learn Python library which is available publicly for download. These methods were chosen over more complex deep-learning methods which often underperform on small tabular datasets (Grinsztajn et al., 2022) and require significant time and expertise for hyper-tuning (Mohammed and Kora 2023), and other complex ensemble methods which can require more computing resources without increased accuracy."

In response to adding a discussion about why certain models performed we added the following into the model discussion

4.1 Probability estimates for chosen models and application to downcore records

4.1.1 Model accuracy

"The F1 score evaluates the accuracy of a model's predictions of both precision (how many predicted positives were positive) and recall (from all the positives, how many positives did the model predict) and can balance between understanding false positives and false negatives (Boozary et al., 2025). This score allows for a more robust accuracy when measuring each model. Many things may explain differences in F1 scores across our models. For example, K-NN, SVM, and CART models are prone to overfitting (Huang et al., 2005; Berk, 2008 Jadjav and Channe, 2013), which may have accounted for their lower F1 scores (Table 1). RF generally does not overfit due to its ability to handle noise in the datasets (Parmar et al., 2019), which may result in a higher F1 score. While LR does not typically overfit, the lower F1 score may be due to its assumptions of linearity (Nick and Campbell 2007), which may be problematic if there is no clear division in the dataset. The balanced versus unbalanced datasets may have also impacted performance. RF generally handles unbalanced datasets well (Anaissi et al., 2013), and the SMOTE dataset only offered marginal improvements to the F1 score, while CART's F1 score was significantly improved with the balanced SMOTE datasets."

2) The study primarily focuses on semi-arid and arid regions, but are the results generalizable to other climate settings? Note that one of the tested sediment records extends back to 35k years BP. Also, there is little discussion of how different environmental conditions might affect the model performance.

Response: Thank you so much for your insight and suggestions. It is true that we focus on arid and semi-arid regions due to the availability of the core records for our analysis. However, this model can indeed be utilized globally, but has not been tested in sediment archives from these contexts. The training dataset, however, is based on a global database, making it applicable globally. To address this, we have added a new section to address the issues with application of the models to sedimentary sequences and removed the words semi-arid and arid environments in the introduction, methods, and conclusion to reflect this insight. We also added a figure in the supplement that shows the distribution of the modern database across the *Köppen*-Geiger climate gradient.

**4.2.4 Considerations for application to paleo-sedimentary sequences**

"The samples from the modern training dataset come from a wide range of modern environmental contexts, making the model applicable to most brGDGT based paleoenvironmental reconstructions. The brGDGT based ML probability outputs on the Padul core, which is 36,000 years old, closely align with the independent pollen and XRF water depth reconstructions during the last glacial period (Figure 8), confirming the model's relevance beyond the Holocene. However, the distribution of modern samples across the Köppen-Geiger climate gradient is not well balanced (Figure S9), with temperate environments well represented, while tropical, arid, and arctic conditions are under-represented. Considering this, caution is advised when applying this model to sedimentary records from these climates. Additional caution is warranted for deep-time records where no modern analogs exist."

[Figure]

Figure S9. *Köppen*-Geiger climate distribution of samples in the brGDGT database created with the kgcpy (Yu et al. 2024) library in Python. O are samples without a label or coordinates assigned.

Citation: Yu, Xuanji, Julian Ascencio, and Roger French. "Open-Source Climate Classification Package: kgcPy." *2024 IEEE 52nd Photovoltaic Specialist Conference (PVSC)*. IEEE, 2024.

3) The results rely heavily on probability estimates to detect mixed-source environments, but their uncertainty and confidence intervals are not entirely clear. How robust are these estimates? How reliable would they remain in cases of overlapping depositional influences (ex. lake vs peat transitions)?

Response: Thank you for this feedback. Yes, we agree that more information is needed on the confidence intervals and the robustness of these estimates. Therefore, we have created 95% confidence intervals by bootstrap 500 times the probabilities on the downcore predictions. We have updated the text to include this information in the methods and discussion section, and what this may mean for applying our model to other sedimentary sequences and in mixed contexts. We have also updated Figure 4 to include these confidence intervals:

Text added:

Methods: "To estimate the 95% confidence intervals for each downcore record, we performed 500 bootstrap resampling on the probability predictions. These were computed separately for each record to reflect their individual variance.:"

Updated draft figure:

[Figure]

**Figure 4:** Downcore probability estimates with 95% confidence intervals and changepoint breaks from Random Forests (RF) on the SMOTE dataset with a sigmoid calibration. Results from the Padul (1) and Vanevan (2) records are broken down by lake probabilities (blue curves - a), peat probabilities (green curves -b), and soil probabilities (brown curves - c). Highlighted grey and white boxes indicate changepoint mean breaks identifying phases. Probability estimates from other models can be found in Supplement 1. (Fig. S5 – S8)

Text added:

4.2.4 Considerations for application to paleo-sedimentary sequences

"The log-loss score of 0.31 for the Random Forest model with a sigmoid calibration indicates robust performance in detecting provenance shifts, including sequences characterized by mixed sources. Nonetheless, the associated 95% confidence intervals exhibit variability along the sedimentary profiles, requiring caution when interpreting the model's output. Furthermore, given that the modern reference dataset is contingent on the accurate classification of the sedimentary context, as reported by original authors, and does not account for mixed provenance in the modern database (e.g., lake sample with high amounts of soil inputs) uncertainties in provenance classification remain possible despite the model's accuracy."

4) The study validates machine learning results using pollen and NPP data, but this comparison has uncertainties among which are variations in pollen/spore productivity and dispersal over time. These uncertainties could also affect the reliability of pollen-based reconstructions. A more extended comparison with established proxies of local relevance (e.g., geochemical elemental data, stable isotopes) would strengthen the argument that machine learning provides superior brGDGT provenance detection.

Response: While we understand the concerns with pollen productive and dispersal overtime, we disagree with the reviewer that pollen and NPPs are not relevant local proxy to measure changes in brGDGT provenance. Pollen dispersal and productivity are indeed an issue for pollen records and like all paleo-proxies are imperfect, but the water-depth estimates from these lakes are based on aquatic pollen and NPPs that are considered a local proxy. Both of these water-depth reconstructions were previously peer-reviewed in these separate studies done by separate non-coordinated researcher teams showing the accepted reliability of these reconstructions in the pollen communities. Most of the pollen and NPPs utilized in for these studies comes from aquatic taxa or taxa that is local to the lake, including Cypereaceae and Typha. The NPPs include aquatic algae, such as *Pediastrum* and *Botryococcus* that live in the lake. In addition, the spores utilized in these records are also of local origin. In addition, many gdgt articles have utilized pollen and NPPs to verify their GDGT results (see text below). Both cores, however, also had XRF analysis done which corresponds to the pollen water depth estimates and our datasets. **We have provided an extended discussion in the manuscript including the XRF data from environments and updated figures (6 and 7).**

We have added additional text to explain the validation.

Introduction:

This section now reads: "Aquatic pollen and NPPs has previously been used to verify changes in provenance in brGDGT communities from fossil records (i.e., Robles et al., 2022; d'Oliveira et al. 2023; Ramos Román et al., 2022; Barhoumi et al., 2023). In addition, we also compare our results with XRF core scanning data from the same sedimentary sequence to strengthen the analysis."

Methods section now reads:

"Although pollen and fungal spore dispersal can be an issue, the fossils of semi-aquatic plants, fungal and fern spores, along with algae should come primarily from around and inside the basin (Gelorini et al., 2013; Gill et al., 2013) which reflects their usage as local indicators of change. Percentages of the aquatic and NPP taxa were calculated by summing all relevant pollen types for each record and dividing each taxon by the total sum. We calculated and re-calculated key brGDGT-based indices (Table 1) to compare our machine learning results with the brGDGT record as well as the aquatic pollen and NPPs. In addition, we also compared our results with the principal components output on the XRF datasets, also taken from the same cores, published in Robles et al. (2022) and Camuera et al. (2018). The descriptions of this analysis can be found in the original publications."

Citation added:

Gill, Jacquelyn L., et al. "Linking abundances of the dung fungus Sporormiella to the density of bison: implications for assessing grazing by megaherbivores in palaeorecords." *Journal of Ecology* 101.5 (2013): 1125-1136

Gelorini, Vanessa, Immaculate Ssemmanda, and Dirk Verschuren. "Validation of non-pollen palynomorphs as paleoenvironmental indicators in tropical Africa: Contrasting~ 200-year paleolimnological records of climate change and human impact." *Review of Palaeobotany and Palynology* 186 (2012): 90-101.

This section now reads:

"PCA analysis was also conducted on the XRF dataset on both the Padul and Vanevan cores, and the authors used it as a proxy for lake-level change. For Padul Camerua et al., (2018) attributed negative loadings of PCA-1 with higher lake levels (Ca, Sr, Si, A, MS) and positive loadings with lower (Fe, S, Br, TOC, C/N). For Vanevan, Robles et al., (2022) associate higher lake levels with the PCA-1 and higher positive loadings (P, K, Al, Mg, Si, Ti, Fe) and negative (S) with lower lake levels. The probability estimates in the Padul record exhibit trends analogous to the Vanevan results, with an alignment with water depth as indicated by pollen, NPPs, and XRF (Fig. 7B). The estimates derive from the pollen data and XRF data from Camiera et al., (2018) indicating a low water stand in phase 4, a high water stand in phase 3, a fluctuating high to low to high stand in phase 2, and a high, fluctuating to low to high stand in phase 1. The observed trends are reflected in our brGDGT-based ML lake probability estimates. Similar to the Vanevan record, the Padul record predominantly features samples with brGDGT-based ML lake probabilities assigned to lakes. However, there is greater variation among categorical types. This is evident in phases 4 and 3, where peat and soil probabilities are combined with lake probabilities, and in phase 1, where notable fluctuations occur in soil and lake probabilities."

Draft figures (these figures will be cleaned up prior to the manuscript being resubmitted, if accepted, but due to time we wanted to show that the PCA from the XRF data B corresponds to the waterdepth and brGDGTs).

**Vanevan**

[Figure]

[Figure]

Minor points

Introduction:

Could the authors elaborate more on the specific limitations of past methods in br-GDGT provenance detection?

Response: Thank you for this suggestion. We have gone ahead and added the following text in the introduction to give a brief history of past brGDGT provenance. This section now reads:

"As climatic or successional changes occur concurrently with temperature variations, isolating the effects of source changes on the MBT'$_{5ME}$ is challenging. Several indexes and ratios have been developed to detect brGDGT provenance change. The BIT index (Hopmans et al., 2004), and later the IIIa/IIa ratio (Xiao et al., 2016), for example, were designed to identify terrestrial organic input in marine sediments (Hopmans et al., 2004). Although useful in marine contexts, these indexes have had limited success in lacustrine terrestrial environments (e.g., Martin et al., 2020). Ternary diagrams are commonly used to visualize brGDGT (e.g., Russell., et al. 2018; Naafs et al.), enabling the comparison between fossil and modern datasets. These diagrams, however, reduce the data size to three variables, limiting their usefulness in isolating the influence of provenance change on the individual brGDGT isomers. Recently Martínez-Sosa et al., (2023), employed supervised machine learning (ML) to identify changes in

provenance using classification models based on modern samples. Their success highlights the power ML applications can have in solving difficult issues. ML applications differ from traditional statistics applications by focusing on prediction rather than inference (Bzdok et al., 2018). ML's power over these conventional methods lies in their ability to handle data with multiple variables for a few subjects while examining non-linear relationships within the datasets (Bzdok et al., 2018). Martínez-Sosa et al. (2023) models proved effective at identifying shifts in provenance; a limitation of their study, however, is the inability to detect periods of mixed provenance. This paper presents a strategy for identifying provenance changes across lacustrine, peat, and soil depositional environments, including mixed contexts, utilizing a new global brGDGT database, machine learning techniques, as well as environmental reconstructions based on pollen, non-pollen palynomorphs, and XRF datasets."

The discussion on machine learning techniques is somewhat broad; more details on how these approaches differ from traditional statistical methods would be very helpful.

>    Response: thank you for this response we have added the following sentence into the introduction, as seen integrated into the paragraph above.

"ML applications differ from traditional statistics applications by focusing on prediction rather than inference (Bzdok et al., 2018). ML's power over these conventional methods lies in their ability to handle data with multiple variables for a few subjects while examining non-linear relationships within the datasets (Bzdok et al., 2018)."

Material and Methods:

- The manuscript states that models were trained and tested using a new modern database, but it does not provide sufficient details on data preprocessing and splitting strategies. E.g. What were the rationale behind dataset division (train-validation-test)?

Response: In order to test the machine learning we decided on a 60:20:20 split for training, validation, and testing based on the best practices. We initially tested these models with an 80:10:10 data split but did not find any improvement and found that our initial split was enough to provide data for training, validation, and testing. We have added the following to the text:

"The data was split into a 60:20:20 training, testing, and validation set. This provided enough data to train the model with high accuracy and ensure that testing and calibration could occur on datasets that were previously unseen during training."

The use of SMOTE seems suitable for handling class imbalance, but was there any testing to ensure it did not introduce artificial biases?

Response: SMOTE provided marginal gains in both the accuracy of the classification of the model and and probability estimates over the raw unbalanced datasets. However, testing occurred with both datasets to make sure that bias did not occur. The full probability models from both are in the supplement figure (S7 and S8) for comparison along with table 1.

In addition, we analyzed the distribution and used a Kolmogorov-Smirnov test comparing the original and smote datasets. These results show that bias was not introduced with smote. We have added the following text into the document:

Text added: lines 56-59 The distribution of the SMOTE samples and original database were plotted and a principal component analysis and Kolmogorov–Smirnov test were run to verify that no bias was introduced (results in supplement Figure S8 & S9 and Table S1).

Figures added:

[Figure]

Figure S9. Distribution of brGDGTs between the original and smote datasets. Overlapping distributions suggests that bias was not introduced with the SMOTE samples.

[Figure]

Figure S10. PCA distribution of the original and SMOTE datasets

| | | |
|---|---|---|
| A. | Original class distribution: | |
| | SampleTypeNum | |
| | 1 0.507672 | |
| | 0 0.259097 | |
| | 2 0.233231 | |
| | Name: proportion | dtype: float64 |
| | | |
| | SMOTE class distribution: | |
| | SampleTypeNum | |
| | 0 0.333333 | |
| | 1 0.333333 | |
| | 2 0.333333 | |
| | Name: proportion | dtype: float64 |
| | | |
| B. | Kolmogorov-Smirnov Test between original and SMOTE datasets (for each feature): | |
| | Ia: KS statistic = 0.0323 | p = 1.8211e-01 |
| | Ib: KS statistic = 0.0277 | p = 3.4250e-01 |
| | Ic: KS statistic = 0.0401 | p = 5.0102e-02 |
| | IIa_5me: KS statistic = 0.0300 | p = 2.5261e-01 |
| | IIa_6me: KS statistic = 0.0742 | p = 6.9874e-06 |
| | IIb_5me: KS statistic = 0.0161 | p = 9.2414e-01 |
| | IIb_6me: KS statistic = 0.0525 | p = 3.6958e-03 |
| | IIc_5me: KS statistic = 0.0658 | p = 1.0150e-04 |
| | IIc_6me: KS statistic = 0.0094 | p = 9.9994e-01 |
| | IIIa_5me: KS statistic = 0.0297 | p = 2.6297e-01 |
| | IIIa_6me: KS statistic = 0.0555 | p = 1.7547e-03 |
| | IIIb_5me: KS statistic = 0.0315 | p = 2.0594e-01 |
| | IIIb_6me: KS statistic = 0.0129 | p = 9.9015e-01 |
| | IIIc_5me: KS statistic = 0.0616 | p = 3.4764e-04 |
| | IIIc_6me: KS statistic = 0.0190 | p = 7.9730e-01 |
| | | |

Table S1. A counts of each class type in the dataset B. Kolmogorov-Smirnov Test between original and SMOTE datasets (for each brGDGT feature in the dataset)

Results:

The results section provides a detailed evaluation of model performance, but it lacks clarity on what is a meaningful improvement in classification accuracy. For example, what is the practical significance of a 0.72 vs. 0.90 F1 score in this context?

Response we have added an extended discussion on F1 accuracy and model comparison in the discussion (see above). This section now reads:

"The F1 score evaluates the accuracy of a model's predictions of both precision (how many predicted positives were positive) and recall (from all the positives, how many positives did the model predict) and can balance between understanding false positives and false negatives (Boozary et al., 2025). This score allows for a more robust accuracy when measuring each model. Many things may explain differences in F1 scores across our models. For example, K-NN, SVM, and CART models are prone to overfitting (Huang et al., 2005; Berk, 2008 Jadjav and Channe, 2013), which may have accounted for their lower F1 scores (Table 1). RF generally does not overfit due to its ability to handle noise in the datasets (Parmar et al., 2019), which may result in a higher F1 score. While LR does not typically overfit, the lower F1 score may be due to its assumptions of linearity (Nick and Campbell 2007), which may be problematic if there is no clear division in the dataset. The balanced versus unbalanced datasets may have also impacted performance. RF generally handles unbalanced datasets well (Anaissi et al., 2013), and the SMOTE dataset only offered marginal improvements to the F1 score, while CART's F1 score was significantly improved with the balanced SMOTE datasets."

The comparison of sigmoid and isotonic calibration functions is interesting, but it is unclear why certain calibrations improved some models but worsened others. More discussion on the underlying reasons for these differences is needed (at least these could be included in the supplementary material).

Response: Thank you for this recommendation, the following has been added into the text under the discussion results

"For the logloss scores, logistic regression is already calibrated (Kull et al., 2017a) so calibrating may result in a lower log loss score, with both a sigmoid and isotonic calibration. SVM does not produce true probabilities by default and needs to be calibrated for these results (Kull et al., 2017b). By calibrating them with a sigmoid or isotonic regression, the output turns to true probabilities which may result in a lower log loss score.  For K-NN, CART, and RF calibration improved the models on both datasets."

Discussion:

The proposed "brGDGT wetland index" is an interesting addition, but more validation is required. The assumptions behind the brGDGT wetland index appear to be valid in modern datasets, but their applicability to fossil records is problematic mainly due to the fact that pollen productivity and dispersal (incl. source area) vary over time due to climatic, ecological, and taphonomic factors (incl. differential preservation). Because of these reasons, a water level reconstruction based solely on pollen/spores may also be problematic.

Response: We agree with both reviewer 1 and 2 that the WI needs more data to validate. We have decided to remove this index from this paper at this time and hope to publish it elsewhere at a later date. This section has now been replaced with a new section 4.2.4 Considerations for application to paleo-sedimentary sequences

Added citations:

Anaissi, Ali, et al. "A balanced iterative random forest for gene selection from microarray data." *BMC bioinformatics* 14 (2013): 1-10.

Berk, Richard A. "Support vector machines." *Statistical Learning from a Regression Perspective* (2008): 1-28.

Bzdok, D., Altman, N. & Krzywinski, M. Statistics versus machine learning. *Nat Methods* **15**, 233–234 (2018). https://doi.org/10.1038/nmeth.4642

Grinsztajn, Léo, Edouard Oyallon, and Gaël Varoquaux. "Why do tree-based models still outperform deep learning on typical tabular data?." *Advances in neural information processing systems* 35 (2022): 507-520.

Huang, Kaizhu, et al. "Local learning vs. global learning: An introduction to maxi-min margin machine." *Support vector machines: theory and applications* (2005): 113-131.

Jadhav, Sayali D., and H. P. Channe. "Comparative study of K-NN, naive Bayes and decision tree classification techniques." *International Journal of Science and Research (IJSR)* 5.1 (2016): 1842-1845.

Kull, Meelis, Telmo Silva Filho, and Peter Flach. "Beta calibration: a well-founded and easily implemented improvement on logistic calibration for binary classifiers." *Artificial intelligence and statistics*. PMLR, 2017a.

Kull, Meelis, Telmo M. Silva Filho, and Peter Flach. "Beyond sigmoids: How to obtain well-calibrated probabilities from binary classifiers with beta calibration." (2017)b: 5052-5080.

Nick, T.G., Campbell, K.M. (2007). Logistic Regression. In: Ambrosius, W.T. (eds) Topics in Biostatistics. Methods in Molecular Biology™, vol 404. Humana Press. https://doi.org/10.1007/978-1-59745-530-5_14

Parmar, Aakash, Rakesh Katariya, and Vatsal Patel. "A review on random forest: An ensemble classifier." *International conference on intelligent data communication technologies and internet of things*. Cham: Springer International Publishing, 2018.-65.

Wang, Xin, Hao Helen Zhang, and Yichao Wu. "Multiclass probability estimation with support vector machines." *Journal of Computational and Graphical Statistics* 28.3 (2019): 586-595